# Role of Mitochondria in Radiation Responses: Epigenetic, Metabolic, and Signaling Impacts

**DOI:** 10.3390/ijms222011047

**Published:** 2021-10-13

**Authors:** Dietrich Averbeck, Claire Rodriguez-Lafrasse

**Affiliations:** 1Laboratory of Cellular and Molecular Radiobiology, PRISME, UMR CNRS 5822/IN2P3, IP2I, Lyon-Sud Medical School, University Lyon 1, 69921 Oullins, France; Claire.Rodriguez-Lafrasse@univ-lyon1.fr; 2Department of Biochemistry and Molecular Biology, Lyon-Sud Hospital, Hospices Civils de Lyon, 69310 Pierre-Bénite, France

**Keywords:** ionizing radiation, high LET particle radiation, carbon ions, radiation quality, ROS, mitochondria, epigenetics, metabolism, signaling, DNA damage response (DDR), hormesis, adaptive response, apoptosis, non-targeted effects (NTEs), genomic instability, cancer, innate and adaptive immune responses, radiotherapy, carbon-ion radiotherapy (CIRT)

## Abstract

Until recently, radiation effects have been considered to be mainly due to nuclear DNA damage and their management by repair mechanisms. However, molecular biology studies reveal that the outcomes of exposures to ionizing radiation (IR) highly depend on activation and regulation through other molecular components of organelles that determine cell survival and proliferation capacities. As typical epigenetic-regulated organelles and central power stations of cells, mitochondria play an important pivotal role in those responses. They direct cellular metabolism, energy supply and homeostasis as well as radiation-induced signaling, cell death, and immunological responses. This review is focused on how energy, dose and quality of IR affect mitochondria-dependent epigenetic and functional control at the cellular and tissue level. Low-dose radiation effects on mitochondria appear to be associated with epigenetic and non-targeted effects involved in genomic instability and adaptive responses, whereas high-dose radiation effects (>1 Gy) concern therapeutic effects of radiation and long-term outcomes involving mitochondria-mediated innate and adaptive immune responses. Both effects depend on radiation quality. For example, the increased efficacy of high linear energy transfer particle radiotherapy, e.g., C-ion radiotherapy, relies on the reduction of anastasis, enhanced mitochondria-mediated apoptosis and immunogenic (antitumor) responses.

## 1. Introduction

The evolution of living matter has been accompanied by different kinds of radiation including ionizing radiation (IR), cosmic radiation, electromagnetic and particle radiation from the sun [1,2,3]. IRs of different quality have been very useful for the discovery of molecular structures and mechanisms of living organisms because IR-induced molecular disturbances often revealed the existence of hidden cellular mechanisms, concerning radiation-induced damage and repair, human health effects in radioprotection and medical applications. IRs affect all cell constituents. Up to now, most studies in radiation biology were centered on the genotoxic effects of IRs involving nuclear DNA (nDNA), and molecular mechanisms of the induction and signaling of DNA damage, DNA damage regulation and DNA repair [4,5,6,7].

More recently, molecular biology studies revealed cellular networks of metabolism homeostasis, growth and proliferation perturbed by radiation. IR interferes with genetic control mechanisms involving nDNA and also with epigenetic control via mitochondria affecting transcription, translation as well as posttranscriptional and posttranslational regulation of gene expression and protein activation.

IR breaks molecular bonds, and lesions are induced either via direct energy deposition and/or via free radicals, i.e., reactive oxygen species, ROS, such as O_2_^•−^ and ^•^OH generated through water hydrolysis [8,9,10,11]. All cell components including nuclear and mitochondrial DNA (mtDNA), lipids and proteins can be affected [12]. The extent and severity of the damage induced depend on radiation quality, radiation energy and the absorbed dose [8]. Low linear energy transfer (LET) IRs (photons, X- and γ-rays) are usually less damaging than high-LET particle radiations (protons, α-particules and heavy ions). Low-LET radiations induce oxidative base damage, single- and double-strand breaks and a few clustered DNA lesions, whereas high-LET IRs induce complex clustered DNA lesions (i.e., locally multiply damaged sites (LMDS)) [13,14].

Various types of DNA damage have been identified and the corresponding enzymatic repair pathways elucidated (reviewed in [4]). In DNA, mismatched bases (due to replication errors) are taken care of by mismatch repair (MMR), oxidative base damage and single-strand breaks (SSBs) by base excision repair (BER), bulky lesions by nucleotide excision repair (NER), whereas the repair of DNA double-strand breaks (DSBs) and clustered DNA damage involves either homologous recombination (HR) or nonhomologous end-joining (NHEJ) repair. An important breakthrough has been the discovery that DNA repair has to be turned on via specific signaling of DNA damage and prior activation of components of the so-called DNA Damage Response (DDR) [5,6,7,15].

Very importantly, mitochondria provide the energy necessary for maintaining cellular integrity and functions. They dominate cell metabolism, bioenergetics and signaling. The mtDNA shows typical epigenetic control and inheritance and does not follow nuclear genomic inheritance, although also subject to nDNA control [16,17,18,19,20]. Thus, mitochondria play an important role in maintaining cellular homeostasis and genomic stability also after IR. The present review highlights recent findings on the implication of mitochondria in epigenetic control, metabolism and signaling, as well as the biological consequences of IR exposures for radioprotection and radiation therapy. Special emphasis is given to the effects of radiation quality comparing low-LET IR with high-LET heavy-ion particle (C-ion) exposures.

## 2. Mitochondria Structure and Function in Normal and Cancer Cells

Mitochondria are small double-membraned organelles in the cytoplasm of eukaryotic cells that function as the main oxygen-consuming power plants [21]. Mitochondria are the principal source of cellular energy through the tricarboxylic acid (TCA) cycle coupled with the electron transport chain (ETC) and oxidative phosphorylation (OXPHOS) [9,22]. One counts approximately 2–10 copies of the mitochondrial genome per mitochondrion, and tens to hundreds of mitochondria per cell. One eukaryotic cell may contain up to several thousand mitochondrial genomes [23].

Mitochondria ensure general cellular metabolism and high energy provision in the form of ATP via OXPHOS to maintain cell survival and homeostasis. Depending on cell type, they may constitute about 4% to 25% of the mammalian cell volume [9] and thus constitute non-negligible targets for IR [16] Mitochondria are thought to originate from the engulfment of symbiotic bacteria into the first cell ancestors [24,25]. Thus, they function quite independently from the nDNA replication, transcription and translation machinery [26,27] and exhibit high reactivity in terms of antioxidant and immune defenses against environmental insults such as toxic physical, chemical and infectious particles. Although tightly embedded in the cellular metabolic network [28], mitochondria are dotted with important quality control pathways such as replication and repair of mtDNA, dynamic fission and fusion mechanisms, protection by antioxidants and free radical scavenging as well as by the removal of dysfunctional mitochondria via mitophagy [21,29]. Tissues with high energy (ATP) consumption exhibit a higher number of mitochondria and mitochondrial mass [30]. For example, the brain shows a high density of mitochondria ensuring energy production by OXPHOS [31,32,33].

### 2.1. Mitochondrial Metabolism and ROS Generation

The production of energy by mitochondria in the form of ATP is of primary importance. ATP is generated through mitochondrial redox processes via fatty acid β-oxidation (FAO) and OXPHOS [34]. The mitochondrial ETC and OXPHOS system consists of five complexes situated in the inner mitochondrial membrane (IMM): complex I (NADH ubiquinone oxidoreductases), complex II (succinate ubiquinone oxidoreductases), complex III (ubiquinone-cytochrome c reductase), complex IV (cytochrome c oxidase) and complex V (ATP synthase) [35]. The flux of protons at these sites activates the ATP synthase (complex V) leading to ATP generation in the mitochondrial matrix.

Although mitochondria are equipped with an outer and inner membrane for protection, during OXPHOS, 1% to 5% of electrons may leak out. Further reactions with oxygen cause the formation of ROS such as superoxide radicals (O_2_^•−^) and subsequently, hydroxyl radicals (^•^OH) and peroxynitrite anions (ONOO^−^) [9,22,36,37,38,39]. About 2–5% of oxygen consumed by mitochondria may give rise to these ROS [36], which lead to oxidative stress by affecting nuclear and mtDNA and all cellular constituents [40]. Increased generation of ROS has been found associated with hypoxia, induction of mutations, cell transformation and death [36].

The environment of the mitochondrial matrix contains two-fold higher ROS levels than the cytosol. Thus, mitochondria are equipped with antioxidant defenses (glutathione peroxidase (GPX), superoxide dismutase (Mn-SOD), which induces transformation of O_2_^•−^ into hydrogen peroxide (H_2_O_2_) and catalase, which catalyzes the inactivation of H_2_O_2_), and a specific battery of DNA repair systems [41]. High ROS levels activate specific antioxidant pathways such as the Nrf2-KEAP1 pathway. ROS change the protein structure of Keap1 that leads to the cytoplasmic release of Nrf2 (NF-E2-related factor-2). Nrf2 is then translocated into the nucleus and induces the transcription of genes involved in antioxidant defenses [41,42]. Nrf2 can also be activated by the stress-responsive gene p62/SQSTM1 (sequestosome 1) [29,30,31,32,33,34,35,36,37,38,39,40,41,42,43]. All these antioxidant defenses are very important to cope with IR-induced free oxidative radicals and ROS.

In addition, the endoplasmic reticulum (ER) constitutes another source of ROS production. In the ER, ROS is also generated in the form of H_2_O_2_ that can constitute about 25% unfolded protein response (UPR) of total cellular ROS produced [10,11,12,13,14,15,16,17,18,19,20,21,22,23,24,25,26,27,28,29,30,31,32,33,34,35,36,37,38,39,40,41,42,43,44].

Excessive ROS production gives rise to mitochondrial dysfunction [35]. ROS also stimulate mitochondria-mediated intra- and extracellular signaling and immune responses [37]. Therefore, mitochondrial redox hubs may constitute very promising targets in anticancer therapy [22].

### 2.2. Mitochondrial Dynamics

Mitochondrial functions include adaptive and maladaptive cellular stress responses [45]. Accordingly, mitochondrial morphology undergoes dynamic changes including membrane fission and fusion regulated by mitochondrial metabolism, respiration and oxidative stress [41,46,47]. In part, mitochondrial dynamics and biogenesis rely on the import of nuclear-encoded proteins, fusion and fission, and they include internal (e.g., ROS) and external (e.g., estrogen) stimuli [47].

Membrane fission involves the cytosolic dynamin-related protein-1 (Drp1), (GTPase dynamin-related protein-1 that has to be activated by phosphorylation at Ser-616) and the mitochondrial fission protein 1 (FIS1) located in the mitochondrial outer membrane (MOM). The mitochondrial fission factor MFF (essential for DRP1 recruitment) and the GTPase DYN2 are also involved in the process. Drp1 binding to receptors (such as MFF, MID49, MID51 and FOS1) forms Drp1 oligomers that cause fission [47]. Fission also has a function in mitochondria quality control. It serves to separate damaged and dysfunctional mitochondrial proteins from the mitochondrial pool [45]. This may include ER-mediated processes such as the mitochondrial UPR removing damaged and misfolded mitochondrial proteins [48].

Mitochondrial fusion promotes the mixing of intact and partially dysfunctional mitochondria to regain fully functional mitochondria from “unhealthy” ones [45]. It involves the mitofusins MFN1 and MFN2 (located in the MOM), and also the optic atrophy factor 1 (Opa1) (located in the IMM) [45]. Fusion of mitochondria is promoted by cardiolipin in the inner mitochondrial membrane through conversion to phosphatic acid by Mitophospholipase D (PLD) [47]. When MFN1 is depleted, small vesicular mitochondria are formed, dispersed in the cell, whereas lack of MFN2 leads to larger vesicular mitochondria in the cell around the nucleus [41]. Interestingly, the growth of cancer cells is impaired by Drp1 knockdown or Mfn2 overexpression because of the restoration of fused mitochondrial networks. Drp1 expression can be associated with migratory cancer cell phenotypes and metastasis [46].

### 2.3. Mitochondrial Genome

The human mitochondrial genome consists of a small, double-stranded circular DNA molecule with 16,569 base pairs that encode 37 genes. Among these, 13 genes code for protein subunits (proteins) of the ETC and OXPHOS system, the 24 other genes code for two ribosomal RNAs and 22 tRNAs involved in the synthesis of the latter [26,49]. The majority of proteins is encoded by >1000 genes of the nuclear genome and required for the replication of mtDNA, its transcription and repair [50]. Most of these nuclear gene products are imported into mitochondria via a specific mitochondrial targeting sequence [51].

Each mitochondrion can contain multiple copies (2–10) of mtDNA [23,24,25,26,27,28,29,30,31,32,33,34,35,36,37,38,39,40,41,42,43,44,45,46,47,48]. Upon DNA damage induction, there is close cross talk between the nucleus and mitochondria, and many nuclear gene products are translocated into mitochondria to repair mtDNA damage and to ensure maintenance of the mitochondrial genome [52,53]. Mitonuclear communication [54], i.e., anterograde nuclear-mitochondrial signaling, depends on the transcription and translocation of gene products involved in mitochondrial biogenesis, whereas retrograde signaling (mitochondrial-nuclear) occurs after mitochondrial stress and involves changes in ROS, mitochondrial membrane potential (MMP), cytosolic Ca^2+^ as well as NADH, H^+^/NAD^+^ and ATP/ADP ratios [48].

As indicated by Kam and Banati, mtDNA accounts for about 0.25% of nDNA (i.e., 16,564 base pairs), and its protein-coding capacity is only about 1% of that of total cellular DNA [55]. Thus, mitochondrial coding DNA sequences are much less directly affected by IR than nDNA. However, mtDNA is more vulnerable than nDNA. In contrast to nDNA, mtDNA has no introns, exhibits maternal inheritance and lacks protective histones [56,57]. mtDNA is impacted in the form of nucleoid-like structures with the mitochondrial transcription factor A (TFAM) condensing mtDNA. TFAM is quite abundant and can cover and stabilize mtDNA. Other proteins can take part in this complexation with mtDNA [58] as well, thus providing some protection against oxidative damage [59,60].

#### Mitochondrial DNA Repair and Mutation

There is less efficient repair in mtDNA than in nDNA. The repair of mtDNA includes mainly BER components of MMR and alternative NHEJ repair pathways coping with mtDNA damage [61,62,63]. The canonical-NHEJ (c-NHEJ) pathway is undetectable but microhomology-mediated end joining (MMEJ) may perform some DSB repair in mtDNA [48,49,50,51,52,53,54,55,56,57,58,59,60,61,62,63,64]. A recent review [65] confirms that the repair systems of BER, MMR and NHEJ are operating in human cell mitochondria. However, the involvement of HR in the repair of DSBs in mtDNA is still questioned. Apparently, mtDNA DSBs are repaired independently of the nDNA repair machinery involving an autonomous mitochondria-specific replication-dependent repair mode involving the mitochondrial replisome (Twinkle and mtSSB) (see [66]). Unrepaired mtDNA fragments may leak out of stressed mitochondria and can mediate pro-inflammatory immunological responses [59].

Among the mtDNA damage induced by low-LET IR, quite common is the induction of a big deletion, i.e., the loss of 4977 base pairs (between nucleotides 8470 and 13,446) coding for important genes such as ATPase, NADPH dehydrogenase complex I and cytochrome c oxidase [9,55,67,68]. However, tumor cells generally show fewer deletions than normal cells after IR [9]. According to Phillips et al., the common deletion does not result from homologous or MMEJ of mtDNA DSBs but from a rather unique autonomous mode of replication-dependent repair, probably retained from early endosymbiosis of a bacteriophage ancestor that is independent of the nDNA repair machinery [66]. Apparently, this involves polymerase gamma (PolG), hexameric Twinkle helicase and mitochondrial single-stranded binding protein (mtSSB).

Chen et al., analyzed the induction of mutations by IR in mtDNA of HeLa cells consisting of the large deletion and single-nucleotide polymorphisms (SNPs) [69]. They found temporal fluctuations in mtDNA 4977 base pairs deletion 72 h after C-ion exposure (not present in surviving cells) but significantly more D310 point mutations after C-ion exposure than after X-ray exposure at similar survival levels [69]. Because point mutations in mtDNA are the main drivers of mitochondrial respiratory dysfunction [70], this suggests more severe effects of high-LET C-ions on mitochondria than those produced by low-LET IR (X-rays). Apparently, such mtDNA mutations can be tolerated by the concurrent presence of multiple copies of non-mutated mtDNA in the same cell; however, they are still readily detectable by the innate immune sensors [59].

### 2.4. Role of Mitochondria in Epigenetic Processes

The involvement of mitochondria in epigenetic processes concerns mainly (1) changes in methylation of DNA, (2) alteration of histones in nuclear chromatin and (3) posttranslational gene regulation by noncoding microRNAs (miRNAs) [71,72]. The expression of nuclear as well as mitochondrial genes is epigenetically regulated.

Similar to nDNA, mitochondria are also under epigenetic control (mitoepigenetics) involving mtDNA methylation and noncoding RNAs (ncRNAs) including miRNAs [56,73]. Epigenetic control of mtDNA function is defined as the control of methylation patterns by specific DNA-methyltransferases without changing the DNA sequence. It differs slightly from that of nDNA [16].

During transcription of mtDNA, transcription intermediate nucleic acid species may also be generated such as double-stranded RNAs, uncapped mRNAs and DNA:RNA hybrids as well as TFAM that stimulate immune responses [59].

mtDNA is mostly methylated at CpG dinucleotide islets by DNA methylases (e.g., DMNT1 and DMNT3a, DMNT3b). These methylases methylate cytosine at the 5′ position. Among nuclei-encoded DNA-methyltransferases, DMNT1 is targeted to mitochondria and operates within mitochondria [74]. Mitochondrial DMNT1 (mtDMNT1) is present in the mitochondrial matrix and binds to mtDNA. Under hypoxia or loss of p53 the nuclear-encoded transcription factors NRF1 and PGC1α up-regulate the expression of mtDMNT1 in mitochondria, which leads to altered expression of transcripts from mtDNA and mitochondrial functions.

#### 2.4.1. IR Interferes with Methylation of DNA

Importantly, IR has been shown to provoke global decreases in DNA methylation *in vitro* and *in vivo* [16,75]. Mechanistically speaking, Madugundu et al., have shown that after γ-irradiation of Fischer glioma cells (F98) the induced hydroxy radicals can oxidize 5-methylcytosine in CpG islets in DNA leading to the formation of 5-hydroxymethylcytosine, which blocks the activity of the methylase DNM1 [76]. This causes improper methylation inheritance during mitosis and global hypomethylation. Moreover, ROS can oxidize guanine to 8-oxoG adjacent to cytosine in CpG, inhibiting cytosine methylation. In this way, ROS may contribute to the hypomethylation of DNA [77]. In comparison with nDNA, mtDNA is believed to be mostly hypomethylated [59]. As reported by Cavalcante et al., mtDNA methylation is also linked to the regulation of gene expression and to mtDNA copy number [73]. Up-regulation of DMNTs in mitochondria was found to be related to oxidative stress and apoptosis.

It is well-known that epigenetic alterations such as posttranslational histone modifications and DNA methylation modify gene expression and chromatin structure of nDNA [54]. Importantly, mitochondria-directed biochemical reactions ensure the production of ATP for phosphorylation, acetyl-CoA for acetylation of histones and folate metabolism, which generates s-adensylmethionine (SAM) needed for methylation of DNA and histones [56,78]. Methylations of DNA in promoter regions, i.e., hypermethylation of genes, inhibit transcription and gene expression leading to gene silencing. Demethylation leads to hypomethylation and may cause genomic instability.

Methylation of cytosine in nDNA belongs to the epigenetic (histone) code together with acetylation, ubiquitylation, phosphorylation and acetylation of the N-terminal amino acids of histones [54,79]. All these modifications depend on the cellular energy status because the metabolites involved are regulated through mitochondrial-mediated metabolism and function [54].

The epigenetic modifications are crucial for DNA repair, cell cycle regulation, apoptosis and genomic instability. For instance, several DNA repair genes are found to be nDNA methylated in the promoter region of tumor cells [80].

After IR exposure, the DNA of several tumor cell lines was demethylated [81] and several proteins’ histones were acetylated [82]. Acetylation of histones is regulating global gene expression via products of the mitochondrial TCA cycle. Methylation of histones may activate or repress the transcription of nuclear genes. Conversely, inhibition of methyltransferases promotes mitochondrial biosynthesis via PGC1α and an antioxidant response via Nrf2 (NFE2C2) [54].

According to Szumiel, IR-induced mitochondrial dysfunction which gives rise to highly increased intramitochondrial ROS can cause mutations in mtDNA and also affect the epigenetic control mechanisms of nDNA by reducing methyltransferase activities [16]. This results in mitochondria-mediated hypomethylation of nDNA that can be transmitted to the progeny of irradiated cells. In this respect, it is important to keep in mind that mtDNA is maternally inherited through the oocyte during conception. Paternal mtDNA is destroyed [83].

Indeed, in conditions of redox stress, the expression of mtDMNT1 (nucleus-encoded and mitochondria-targeted) is increased via the transcription factors nuclear respiratory factor 1 (Nrf1) and PDG1α [84]. This suggests that mtDMNT1 may have a regulatory role during mitochondrial oxidative stress. Mitochondrial metabolism supplies the methyl donor SAM, i.e., the donor molecule for methylation by DMNTs. Thus, Kietzmann et al., concluded that the methylation status of mtDNA and nDNA can be affected not only by mitochondrial dysfunction associated with increased ROS production but also by interference with mitochondrial metabolism, i.e., the mitochondrial Krebs cycle [84]. In mice, IR-induced DNA damage and related changes in DNA methylation were found to cause genomic instability and bystander effects in the testis tissue [85].

#### 2.4.2. IR-Induced Epigenetic Changes Depend on Radiation Quality

Goetz et al., noticed that the epigenetic response (including genomic DNA methylation) elicited by IR, although being quite similar after high-LET irradiation (^56^Fe ions) and low-LET irradiation (protons) exposure, is more related to radiation quality and track structure than to LET [86]. Differences in oxidative stress produced were thought to explain the different responses. γ-Irradiation induced a higher level of DNA methylation in resistant head-and-neck squamous cell carcinoma (HNSCC) cells (rSCC-61) than in sensitive HNSCC cells (SCC-61) [87]. DNA methylation was especially elevated in the promotor region of the cyclin CCND2 gene in rSCC-61 cells suggesting that DNA methylation may serve as a marker of radioresistance.

Kennedy et al., confirmed that DNA methylation after IR exposure is dependent on radiation quality [88]. They observed significantly different methylation patterns in human bronchial cells exposed to low- and high-LET irradiation. Low-LET X-irradiation gave rise to methylation in genes and intergenic regions. High-LET ^28^Si-ion irradiation resulted in DNA methylation in repressed chromatin sites of heterochromatin, and ^56^Fe-ion irradiation in DNA hypermethylation in CpG islets of open chromatin regions. The authors concluded that high-LET heavy ions may leave a lasting epigenetic imprint in tumor cells that allows discrimination from normal cells of the same tissue.

An example of an important modulation by epigenetic changes induced by IR exposure is the phosphorylation of the histone variant H2AX induced by IR initiating the DDR signaling cascade involving DNA repair, cell cycle arrest and apoptosis [5].

#### 2.4.3. IR-Induced Epigenetic Effects via microRNAs

Posttranslational epigenetic modifications may involve ncRNAs that can act at transcriptional, posttranscriptional and posttranslational levels [73,89]. Bandiera et al., were the first to demonstrate that in human cells (HeLa cells), addressing of nuclear RNA interference components to the human mitochondria is taking place and that there is an important cross talk between the nucleus and mitochondria via a subset of miRNAs, i.e., small noncoding RNAs (“mitomiRs”) [90].

miRNAs are brought about by RNA polymerase II transcribing miRNA gene cleavage by the RNAse DICER to create a miRNA duplex of 22 base pairs. A single-stranded RNA is obtained via complex formation with an argonaute protein to form the RNA-induced silencing complex (RISC). This complex interacts with the complementary mRNA of a specific gene, which leads to repression of the gene. Thus, miRNAs play an important role in the regulation of gene expression.

The scientific literature on noncoding miRNAs is already abundant (see the review by Marta [91]). In fact, miRNAs modulate the expression of a wide range of post-translational targets. For example, miRNAs can interfere with the very important DDR pathway. For example, DSB repair can be repressed by miRNA-24 by down-regulating the expression of the histone variant H2AX resulting in increased cell death, and miRNA-421 can control the ataxia-telangiectasia-mutated ATM gene that plays a central role in DDR. Both miRNAs affect radiation sensitivity. MiRNA-101 causes IR-induced radiosensitization via DNA-PKcs and ATM. MiRNA-34 is a target for the tumor suppressor gene p53.

At the mitochondrial level, this is of utmost importance: for instance, ATP is generated through OXPHOS by four protein complexes (I–IV) (the ETC that drive ATP production by ATP synthase). miRNAs can modulate the processes of each complex (I–IV) [92]. Thus, these ncRNAs (miRNAs) can control glycolysis, mitochondrial energy status and the expression of mitochondrial metabolism-related genes [93].

Importantly, ncRNAs inside mitochondria can be generated in two ways: ncRNAs produced by the nuclear genome and those produced by the mitochondrial genome. Importantly, both types of ncRNAs were shown to be involved in apoptosis by anterograde and retrograde signals [94,95,96]. Several ncRNAs were found to be implicated in mitochondria-mediated intrinsic and/or extrinsic apoptosis. For example, MEG3 increased apoptosis with the release of cytochrome c due to reduced expression of Bcl-2 and ProC9 proteins [97].

Among the ncRNAs encoded by mtDNA, two long RNAs, ASncmtRNA-1 and ASncmtRNA-2, are involved in apoptosis in mouse and human tumor cell lines [98].

Obviously, the epigenetic control of cell metabolism and differentiation via mitochondrial ncRNAs also applies after IR. Many ncRNAs regulate crucial metabolic pathways and thus, also those involved in response to IR. X-irradiation has been found to modulate the expression of miRNAs in glioblastoma cells with different DNA repair capacities [99], and in X-irradiated cells and bystander cells, significant differences in miRNA expression were observed [100]. In addition, Mao et al., noted changes in miRNA expression after X-irradiation and suggested that miRNA DNp73 could be a potential regulator of genes involved in DDR signaling and repair (ATM, BRCA1) and genes coding for transcription factors such as p53, NF-κB, Myc and E2F [101]. Recently, a rather specific ncRNA, i.e., miRNA-29b-3p, was identified that interfered with the regulation of WISP-1, a Wnt-1 and beta-catenin-responsive oncogene and enhanced radiosensitivity of prostate cancer cells via mitochondrial apoptosis [102]. This oncogene not only plays a role in prostate cancer but also in colon cancer and glioblastoma. Inhibition of WISP-1 facilitated apoptosis via suppression of Bcl-XL and activation of caspase-3 and PARP1. Radiosensitization of prostate cancer cells (LNCaP) was achieved through the targeting of WISP-1 with miR-29b-3p. This clearly indicates that miRNAs can also be implicated in the regulation of IR-induced mitotic-mediated apoptosis. No wonder that Nguyen et al., emphasized the emerging role of miRNAs in the treatment of pancreatic cancer by RT [103].

In addition, differences in profiles of miRNAs after exposures to IR depending on radiation quality have to be considered. For instance, specific miRNA signatures were observed in blood samples taken from total body exposure of mice to 0.1–2 Gy of ^12^C-ion and ^56^Fe-ion irradiation. He et al., noted a change in miRNA profile induced by 2 Gy of C-ion-irradiation in mouse testis [104,105].

Recently, Jin et al., showed that radioresistant non-small cell lung cancer cells A549-R11 and H1299 (NSCLC) are more resistant than parental A549 (NSCLC) cells to low-LET X-irradiation, whereas very resistant A549-R11 cells could be equally radiosensitized by high-LET C-ions (100 MeV/u, LET 50 keV/μm) [106]. Certainly, research on the radiosensitizing effect of specific lnRNAs on tumors merits being actively pursued [107]. This rapidly developing research field is likely to stimulate beneficial applications in radiotherapy [108].

Recently, Jin et al., showed by high-throughput sequencing that other types of ncRNAs, e.g., 40 circRNAs were up-regulated, whereas 184 circRNAs were down-regulated in radioresistant A549-R11 (lung cancer) and H1299 (NSCLC) cells [106]. From this, together with other interactive miRNAs and mRNAs, a circRNA–miRNA–mRNA network was constructed that may serve as a prognostic biomarker for RT and CIRT of NSCLC treatments [106].

#### 2.4.4. IR-Induced Epigenetics and Genomic Instability

A recent review by Baulch retraces the present evidence for links existing between IR-induced epigenetic mechanisms, mitochondrial dysfunction and genomic instability [78]. It goes together with the notion that non-targeted effects (NTE) do not directly involve genomic DNA. In line with the initial findings of Nagasawa and Little [109,110] and Kadhim et al. [111] with α-particle irradiation (^238^Pu, LET: 121 keV/μm) and Sabatier et al. [112] with high-LET neon (86 keV/μm), argon (1207 keV/μm) and lead (13,600 keV/μm) exposures observed that such exposures led to chromosomal imbalances and chromosome instability in hematopoietic cells and human fibroblasts, respectively that were perpetuated for many cell progeny generations and involved fractions of the cell population that were not directly traversed by these heavy particles. Very importantly, genomic instability involved early steps of cell transformation [112] and carcinogenesis [113]. The existence of NTE after α-particle exposure was confirmed by Lorimore et al. [114]. This was further supported by Morgan [115,116,117], showing that non-targeted and delayed effects of IR could induce genomic instability *in vitro* and *in vivo*.

In the following years, the links between genomic instability and epigenetic effects involving mitochondrial dysfunction became more and more evident [78]. Based on early observations, Spitz et al. [118,119] developed a unifying concept implicating mitochondrial activities in NTE and genomic instability. Indeed, as emphasized by Baulch, mitochondria provide important substrates for epigenetic regulation: ATP for phosphorylation and acetyl-CoA for acetylation of histones, the substrate for the methylation of histone and DNA (i.e., S-adenosylmethionine through folate metabolism), and intermediates of the mitochondrial TCA cycle [78]. However, persistent ROS and oxidative stress induced by IR causes mitochondrial dysfunction [16,55,120] and is likely to alter not only the metabolic energy status of cells but also changes of epigenetic regulation hallmarks of carcinogenesis [121]. Indeed, IR-induced mitochondrial dysfunction perpetuated genomic instability [122]. In addition, Dayal et al., suggested that dysfunction of mitochondria and mutations in complex II produced by excessive mitochondrial O_2_^•−^ production could lead to genomic instability [123]. Oxidative stress in cancer cell-associated fibroblasts (CAFs) due to mitochondrial ROS generation was clearly related to genomic instability [124].

Exposures of normal human fibroblasts to fractionated very low-dose X-irradiation (0.01 or 0.05 Gy/fractions) for 31 days induced mitochondrial ROS, disruption of AKT/cyclin D1 cell cycle signaling via the oxidative inactivation of protein phosphatase 2A (PP2A) resulting in nuclear cyclin D1 accumulation, genomic instability and senescence [125,126].

### 2.5. Mitochondria and the Specific Bioenergetics and Metabolism of Cancer Cells

Cancer cells are characterized by genetic and epigenetic changes that provide unusual proliferative, migrative and invasive properties and resistance to tissue and microenvironment control mechanisms [121]. The high proliferation rate of cancer cells is associated with a high level of mitochondria and increased mitochondrial energy metabolism. These are accompanied by profound metabolic changes conferring protection against tissue control mechanisms and often increased resistance to chemical- or radiation-induced insults.

The main mitochondrial hallmarks of cancer cells are deregulated cellular energy metabolism, i.e., use of glycolysis instead of OXPHOS to generate ATP even under aerobic conditions (Warburg effect) [127,128,129,130,131], and other alterations of mitochondrial metabolism together with the capacity to evade cell death (mainly relying on antiapoptotic Bcl-2 functions) [132].

In normal, non-cancer cells, mitochondrial energy production includes the following: (1) ATP is produced through the mitochondrial ETC in which a proton gradient is generated across the inner mitochondrial membrane by the respiratory complexes I, III and IV driving ATP synthesis via complex V (ATP synthase) [133], (2) mitochondrial OXPHOS yields 32 molecules of ATP per glucose molecule, whereas aerobic glycolysis yields only 4 ATP molecules per glucose molecule and anaerobic glycolysis only 2 ATP molecules per glucose molecule [134]. About 91% of cancer cell ATP is produced by mitochondrial OXPHOS in the presence of oxygen but only 36% of ATP under hypoxia [132].

By contrast, cancer cells exhibit, instead of normal oxidative metabolism, mitochondrial hyperactivity with a highly elevated metabolism and anaerobic glycolysis (with reduced ATP production) associated with hypoxia and radioresistance [127,135,136,137].

Indeed, to support high proliferation rates, cancer cells switch their mitochondrial energy metabolism from oxidative metabolism to glycolysis in normoxic/hypoxic conditions [121,138,139,140]. To cope with the reduced energy supply of ATP by aerobic glycolysis in comparison to OXPHOS, the extent of mitochondria is highly increased (by a factor of up to 20) to cope with the elevated energy needs of the highly proliferative cancer cells.

The switching from OXPHOS to glycolysis involves high glucose consumption and lactate production (as first reported by [141]). Lactate production is an indicator of the metabolic state of cells and is rapidly released upon damage to the plasma membrane, and appears to be linked to radio/chemotherapy efficacy [142]. Lactate dehydrogenase A (LDHA) is involved in the reversible conversion of pyruvate to lactate in the presence of NADH, H^+^/NAD^+^. In cancer cells, LDHA is up-regulated and promotes the metabolic switch to anaerobic glycolysis with the generation of lactate. Interestingly, LDHA knockdown or inhibition results in attenuated glycolysis, increased ROS and a decrease in tumor metastasis [143].

Interestingly, reduction of lactate efflux in gliomas (by an inhibitor) caused a decrease in reduced glutathione (GSH) and increased sensitivity to IR [144]. Moreover, lactate dehydrogenase regulates cell migration and invasion by increased lactate output. In fact, lactic acid levels are associated with increased tumor cell motility and migration of cancer cells, and the incidence of distant metastases. This is associated with activation of metalloproteinases (MMPs) and cathepsin, up-regulation of VGEGF, HIF-1α, and TGF-β2 and increased expression of E-cadherin [145,146]. Activation of MMPs such as MMP-2 and MMP-9 may involve translocation to mitochondria and metastasis formation [147].

In most cases, cancer cells exhibit dysfunctional mitochondrial enzymes favoring oncogenic effects, affecting hypoxia-inducible factor 1 (HIF-1α) activation as well as genomic instability (via ROS production and mutations of mtDNA). This may lead to inflammatory processes (via the release of damage-associated molecular patterns (DAMPs) (such as mtDNA, N-formyl peptides and cytochrome c), driving metastasis (through mitochondrial ROS, cytoskeleton and matrix remodeling (involving Src/Pyk2 kinases)) and angiogenesis (involving HIF1α, VEGF and matrix MMPs), and changes in mitochondrial dynamics with fusion and fission [132].

#### 2.5.1. Mitochondrial Hyperactivity of Cancer Cells Can Create DSBs in DNA

Because of the hyperactivity of cancer cells, their mitochondria are unable to keep all radicals produced inside their double-shelled organelles, and some ROS and cytochrome c leak out and induce DSBs in nDNA. Mitochondrial ROS lead to oxidative stress that affects nuclear and mitochondrial DNA and other cellular constituents [40]. Thus, most cancer cells contain a significantly higher level of DSBs than normal cells [148]. These self-inflicted (“spontaneous”) DNA DSBs (spDSBs) in cancer cells are associated with persistent ATM activation and DDR, and more aggressive tumor types. spDSBs are not lethal but can lead to mitochondrial outer membrane permeability, leakage (cytochrome c release, increased ROS production, activation of apoptotic caspases, activation of nucleases (endoG and caspase-dependent DNase (CAD)), persistent nDNA DSBs, initiation of cell death pathways, and in the long run, to genetic instability and oncogenic transformation.

Because of the inherent production of ROS and the lack of robust protective mechanisms, cancer cell mitochondria are particularly susceptible to oxidative stress from other stressors (IR) and sustaining persistent mitochondrial and cellular stress [149].

#### 2.5.2. Epithelial-to-Mesenchymal Transition (EMT) and Metabolic Reprogramming of Cancer Cells

The epithelial-to-mesenchymal transition (EMT) is a characteristic feature of cancer cells associated with metabolic reprogramming. In the process of cell transformation, EMT integrates oncogenic signaling, metabolic transformation and epigenetic reprogramming [150]. EMT is accompanied by profound changes in mitochondrial energy metabolism and vice versa. Alterations of mitochondrial metabolism promote EMT and changes in glucose and lipid metabolism and glutaminolysis metabolism [150].

Mitochondrial dysfunction has been recognized as an important driver of EMT in cancer also stimulating the ability of cancer cells to undergo metastasis formation [151]. However, recently, it has become evident that in conditions of hypoxic cancer, cells may be able to undergo reprogramming, i.e., a shift in mitochondrial energy metabolism from glycolysis back to OXPHOS [152]. This reprogramming enhances energetic and nutritional support for cell proliferation, migration and metastasis formation and chemo- and radioresistance to conventional anticancer treatments. The HIF-1α together with the AMP-activated protein kinase (AMPK) plays an important role in the reprogramming of cancer cells [152]. Metabolic reprogramming involves control of mitochondrial ETC activity and energy production by HIF-1α that interacts with several oncogenes including KRAS, c-Myc, p53 and AMPK together with protein kinase B PKB/Akt, growth factor initiated phosphatidyl-3-kinase (PI3K) and the mammalian target of rapamycin (mTOR) cell signaling pathways. Activation of the heterotrimeric serine/threonine kinase AMPK occurs when the AMP/ATP ratio in mitochondria decreases. This turns on ATP production through glycolysis or OXPHOS. ATP levels in cells switching from anaerobic to aerobic catabolism (oxidative metabolism) are controlled by stimulation of glucose uptake (through activated glucose transporters), aerobic glycolysis and mitochondrial oxidative metabolism. Lactic acidosis may switch cancer cells back from anaerobic glycolysis to dominant OXPHOS [153]. Depending on the microenvironment, different cancer cell lines produced ATP by OXPHOS about twice as much in a lactic acidosis environment (20 mM lactate, pH 6.7), and 6 times less by glycolysis in a non-lactic acidosis environment [153]. Several mitochondria-targeted agents can strongly interfere with mitochondrial energy metabolism in cancer cells [22,139].

Several mitochondria-targeting agents (Mito-metformin, Phenformin, MitoChM, Mito-CP, Me-344 (complex I inhibitors), Mito-Honokiol, Mito-Vitamin-E, MitoQ, MitoTEMPO, and SkQ1) inhibit OXPHOS, and Mito-metformin, Mito-Honokiol, Mito-Vitamin-E, and dichloroacetate) exhibit clear antiproliferative activities in cancer cells [22,139].

Very importantly, mitochondrial reprogramming can reverse the radiation resistance of cancer cells, as recently reported by Sun et al., for glioblastoma cells [154]. These authors showed that these radioresistant cells could be radiosensitized to X-irradiation via mitochondrial reprogramming. By transferring, through endocytosis, normal human astrocytic mitochondria into glioma U87 cells known to be particularly radioresistant, oxidative respiration could be rescued, the Warburg effect attenuated (reversed), malignant proliferation inhibited and mitochondrial apoptosis reactivated. Thus, the radiosensitivity of these glioma cells *in vitro* and *in vivo* (on xenografted U87 tumors in mice) was significantly increased. The NAD^+^–CD38–cADPR–Ca^2+^-transduction pathway appeared to be involved in the endocytosis of the mitochondria [154]. This demonstrates that metabolic switches of mitochondria from OXPHOS to glycolysis sustain tumor cell proliferation and resistance, whereas back switches to oxidative metabolism are strongly tumor radiosensitizing.

#### 2.5.3. Adaptation to Hypoxia and the Role of HIF-1α in Cancer Cells

Most tumors contain hypoxic cells. Mitochondrial anaerobic glycolysis can provide sufficient energy in hypoxic conditions [152,155,156]. For example, in the brain, mitochondria are strongly involved in the adaptation to hypoxic stress [157]. Hypoxia induces a change in respiratory chain function by switching from oxidation of NADH, H^+^ substrate (complex I) to activation of succinate oxidation (complex II) at the mitochondrial level. In other words, hypoxia decreases mitochondrial oxidation and promotes the switching over of tumor cells to glycolysis. Hypoxia activates HIF-1, a heterodimeric DNA-binding protein complex with the constitutive and cytoplasmic inducible subunit alpha [158,159]. It is frequently activated in tumors. In normoxic conditions, it has a short half-life (6 min), whereas, under hypoxia, its proteasomal degradation is prevented [139]. Synthesis of HIF-1α is controlled by MAPK and PI3K signaling systems that are activated by the tyrosine kinase receptor. Agonists are tyrosine hydroxylase, cytokines, growth factors (IGFL) and succinate.

Hypoxia leads to succinate-dependent HIF-1α stabilization, its accumulation, and translocation into the nucleus where it ensures induction of target genes responsible for increased glucose uptake and lactate production, providing long-term adaptation and activation of the succinate-specific receptor G protein-coupled receptor 91 (GPR91, a membrane protein) [157]. This is part of the cellular adaptation to hypoxia. Thus, HIF-1α participates in intercellular systemic interactions and adaptation ensuring energy supply and conferring cellular resistance to IR and chemical drugs. The products of anaerobic glycolysis (lactate) provoke accumulation of HIF-1α in tumor cells initiating the transcription of genes involved in transport, regulation of glycolysis and the pentose phosphate pathway [160]. Therefore, targeting hypoxia and tumor glucose metabolism could be beneficial in the RT of tumor cells [160,161] because hypoxia and nutrient starvation are associated with radioresistance [137].

IR markedly interferes with mitochondrial dynamics, with oxidative metabolism and generation of ROS as well as with the elimination of damaged cell components and dysfunctional mitochondria by autophagy and mitophagy, respectively. Because of the special features of cancer cells in terms of mitochondria-dependent energy metabolism and hyperactivity, the IR-induced changes in cancer cell metabolism are of particular importance with respect to radiotherapeutic treatment outcomes.

## 3. Ionizing Radiation (IR) Effects Involving Mitochondria

In radiation biology, research work has always been largely focused on the damaging effects of IR on the nDNA which are closely related to cell inactivation, induction of mutations, malignant transformation and cancers. IR exposures of mitochondria were thought mainly to affect mitochondrial bioenergetic and biosynthetic metabolism and induce programmed cell death (apoptosis). However, more recently, it has been shown that IR can also affect mitochondrial epigenetic regulation, internal cell signaling and intercellular signaling and communication including bystander effects, inflammatory and immunogenic processes in cell populations, tissues and the whole body via modulation of mitochondrial activity and functions [9,16].

As mentioned above (Section 2.1), because of increased internal ROS levels, cancer cell mitochondria are likely to be more sensitive and subject to dysfunction to additional damage and ROS induction by IR than normal cell mitochondria. Their hyper(re)activity makes them preferential targets in anticancer therapy.

Recently, Grasso et al., isolated a radioresistant cell line from a human SQD9 laryngeal squamous cell carcinoma cancer cell line (HNSCC) after chronic γ-irradiation [162]. The acquired radioresistance of this cell line was found to be associated with more abundant, fitter mitochondria, with increased mtDNA content and a switch from a glycolytic to a more oxidative metabolism. In that case, boosting cancer cell mitochondria content and metabolism can increase radioresistance, which links cancer cell IR responses to mitochondrial functions.

One can say that mitochondria are distinctly involved in IR responses in two ways: (1) low doses (<0.1 Gy) of IR often give rise to nonlinear responses concerning induction of specific genes, activation of proteins, intra- and intercellular signaling including low-dose hormetic and adaptive responses, NTE, genomic instability, cell proliferation, low-dose hypersensitization, and immune (inflammatory versus noninflammatory) responses [163,164], as part of research work of the last 10–20 years involving MELODI and DoReMi initiatives at the European level; and (2) high (>1 Gy) accidental or radiotherapeutic doses lead to linear induction of DNA lesions, mutations, cell malignant transformation, cancer and cell death (apoptosis or even necrosis) as well as effects on the immune system, i.e., inflammatory and/or immunogenic responses. All these effects involve epigenetic control mechanisms and mitochondrial activity and functions.

Low IR doses interfere with cellular homeostasis and elicit early regulatory responses involving mitochondrial control functions on bioenergetics and biosynthesis leading to enhanced antioxidant defenses, increased ATP for DNA repair but also slightly increased mitochondrial ROS generation, mitochondrial fission and fusion, and release of immunologically active cytokines and enhancement of intercellular communication. High IR doses generally induce non-irreversible structural cellular damage within cells. The subsequent struggle for survival after excessive damage relies very much on mitochondria integrity and function mediating longer-term responses such as cell death (apoptosis), inflammation and/or immunogenicity.

### 3.1. Induction of ROS by IR, Mitochondrial Dysfunction and Generation of Mitochondria-Mediated ROS

Low-LET irradiation with photons (X-rays, γ-rays or electrons) and high-LET irradiations such as heavy-ion exposure effectively produce ROS through radiolysis of molecules involving excitation and ionization of water molecules and the production of free radicals (H^•^, HO_2_^•^, ^•^OH) and hydrated electrons (e_aq_^–^) and secondary radicals, H_2_O_2_ and other peroxides [8,9,165,166,167]. Regarding localized cell and tissue responses in cancer therapy, low-LET irradiation by photons gives rise to somewhat homogeneous ionization patterns and radical distribution, whereas heavy-ion irradiation gives rise to inhomogeneous ionization patterns and specific, localized radical distributions along ionization tracks [8,168,169,170].

Mitochondria are the main sources of physiological ROS levels [171,172], which can be strongly enhanced by IR [10,16,173]. Among low-dose IR effects, some authors consider mitochondrial ROS-mediated effects as being even more significant than those mediated by DNA damage [10]. Through the ETC, mitochondria produce ATP and some ROS (O_2_^•−^ radicals) that give rise to other ROS, which can leak out and cause damage to cytoplasmic and nuclear constituents. Because of their high metabolic and proliferative activity, cancer cells show generally increased mitochondria mass and ROS production. This makes them vulnerable to perturbances in cellular metabolism and ROS produced by external genotoxic agents such as IR, which exacerbates cellular ROS and oxidative stress [9]. IR increases the overall load of cell-damaging oxidative radicals that interfere with the mitochondrial redox system, increasing mitochondrial intracellular ROS production and short- and long-term damage to cells and tissues [174]. Therefore, low-LET IR and even more so, high-LET IR, increase ROS-dependent oxidative stress [10,16].

In human U937 cancer cells low-LET IR (γ-irradiation) amplifies ROS and induction of cell death [175]. Kobashigawa et al., reported that low-LET X-irradiation accelerated DRP1-dependent mitochondrial fission leading to delayed mitochondria ROS generation in normal human fibroblast-like cells [176]. Five Gy of irradiation caused an increase in intracellular ROS within minutes that disappeared; however, in 30 min. About 12 h later, the activity of NADH dehydrogenase (mitochondrial Complex I), an enzyme involved in the regulation of the mitochondrial ETC, decreased, indicating mitochondrial dysfunction. This dysfunction was found to be associated with a significant increase of mitochondrial ROS levels and induction of oxidative damage in mtDNA occurring later at 24 h post-irradiation, likely causing genomic instability. Ten Gy of X-rays exposure of human lung carcinoma cells A549 induced a time-dependent increase in mitochondria membrane potential, oxidative respiration, ATP production and increased mitochondrial ETC activity. Cells accumulating in the G2/M phase showed an increase in mitochondria and ROS production. Yamamori et al., confirmed that 10 Gy of X-irradiation can induce additional mitochondrial generation of ROS in human lung cancer A549 cells through up-regulation of mitochondrial electron transport function [177].

In normal skin fibroblasts, Laurent et al., noted that X-irradiation can produce oxidative stress involving the production of superoxide radicals that may lead to highly damaging protein peroxides that are relatively long lived (7 s) [178].

Heavy ions, e.g., C-ion irradiation, can more effectively produce oxygen through water hydrolysis in ionization tracks. The estimated oxygen concentrations produced in tumors are three orders of magnitude higher than those in normally oxygenated and hypoxic regions of unirradiated tumors [166]. C-ions exposure of normal human skin fibroblasts led to oxidative stress (with lipid peroxidation and protein carbonylation) much more significantly than X-irradiation at 24 h post-irradiation [178]. Concomitantly, the levels of antioxidants (SOD, catalase and glutathione peroxidase) were markedly decreased [178]. Seven days after exposure to C-ions, pro-inflammatory IL-6 secretion increased significantly but decreased relatively late compared with X-rays. In addition, Dettmering et al., observed that C-ion irradiation induced higher levels of ROS than X-rays in normal human fibroblasts (AG1522 cells) but the ROS levels induced were not very long-lasting, and, after 10 days, they were back to normal levels when cells started again to proliferate [179]. C-ions were clearly more efficient in ROS production than X-irradiation because a three-fold higher dose of X-rays was needed to obtain the same level of ROS as with C-ions. In human breast cancer cells MCF-7, IR increased ROS generation in a dose-dependent manner [180].

This is in accord with the generally elevated oxidative stress induced by C-ions. Low-LET IR can induce mitochondria-mediated ROS, together with some PARP1 and DNA cleavage and release of nDNA and mtDNA into the cytosol. However, high-LET IR causes more significant increases in ROS production in mitochondria, particularly in cancer cells leading to enhanced cellular oxidative stress and apoptosis.

### 3.2. Low-Dose IR Responses and Signaling Involving Mitochondria

Low doses of IR (<0.1 Gy) tend to affect cell homeostasis by producing hormesis, adaptive responses, low-dose hypersensitivity, as well as bystander and NTE, immunological effects as well as genomic instability [181], whereas radiotherapeutic doses >1 Gy are associated with severe damage, cell cycle arrest, cell death (apoptosis), inflammatory and immunogenic responses. All these effects appear to include changes in epigenetic control mechanisms. Transcriptional profiling corresponds to the genes induced at these different dose levels. In general, at low doses, the genes induced are involved in general metabolism (bioenergetics and biosynthetic functions) whereas high doses activate genes involved in cell cycle arrest, DNA repair and apoptosis [182,183]. Even if these are not yet fully understood, they may well have a bearing on radioprotection and radiotherapeutic issues.

#### 3.2.1. Hormesis

As outlined above, ROS generated by IR or by mitochondria at low, physiologically acceptable levels have signaling and stimulating functions [16,184,185]. Phenomena such as hormesis [186,187,188,189,190] and low-dose adaptive responses as well as innate immune responses can be stimulated.

Slight boosting of mitochondrial metabolic functions by IR increases the actual amount of energy (ATP) available and the initiation of intra- and intercellular signaling by partially damaged or dysfunctional mitochondria should be considered in this context.

In normal cells, IR exposures can indeed stimulate OXPHOS and thus ATP production [9,16,42,177,191]. However, this depends very much on the oxidative and metabolic status of the cells (mass of mitochondria, oxidative stress), and thus is likely to affect normal cells differently from cancer cells, the latter showing already elevated ROS levels and oxidative stress.

Hormetic responses are typical low-dose IR (LDR) responses (<0.1 Gy) including stimulatory effects for cell proliferation and cell survival and up-regulation of antioxidant, DNA repair and immune defenses in vitro as well as in vivo. These effects are beneficial for cells, organs and the whole organism, and may confer increased longevity, embryo production and immunological protection against diseases including infections and cancer [187,192].

Hormesis exhibits some sort of a “wake-up” function to induce adaptive responses to prepare living matter to cope more efficiently with high-dose challenges from IR or other damaging agents. With this, IR-induced hormesis is challenging the so-called linear non-threshold (LNT) hypothesis according to which low-dose IR-induced health risks would be linearly related to low-dose health risks and thus could be assessed directly by linear extrapolation from high-dose effects down to low-dose effects. Indeed, low-dose effects and adaptive metabolic responses would need to be considered also in radioprotection for risk estimations at low doses and low dose rates better to ascertain and rationalize low-dose health risk evaluations [164,187,192].

As outlined by Sthijns et al., the quickest response of cells to oxidative stress is direct enzyme modification, increasing antioxidant GSH levels or inactivating GSH-dependent protective enzymes [193]. Several hours thereafter, a hormetic response is seen at the transcriptional level through up-regulating Nrf2-mediated expression of enzymes involved in GSH synthesis. In the long run, adaptations may occur at the epigenetic and genomic levels.

In fact, following induction of IR-induced oxidative stress, several antioxidant genes, e.g., GPx, Trx, CAT and SOD2 (Mn-SOD) are regulated by Nrf2 after translocation from the cytoplasm to the nucleus [194]. There is evidence that not only activation of the transcription factor NRF2 and glutathione synthesis (leading to cellular protection) but also DNA methylation and epigenetic effects, and the induction of ncRNAs (miRNAs) are involved in hormesis [195].

A meta-analysis of studies in mice revealed that IR doses between 0 and 100 mGy result in protection, and higher doses were found more likely to cause damage [196,197]. Sies and Feinendegen reported that low-dose IR (<100 mGy) effectively induces ROS (including H_2_O_2_) and turns on molecular redox master switches such as Nrf2-Keap or NF-κB to produce hormetic and radioadaptive effects [198,199].

On the other hand, low doses may also stimulate immunogenic antitumor responses involving activation of macrophages and natural killer (NK) cells and innate and adaptive immune responses dependent on mitochondrial activities [200]. Low doses of IR can indeed boost the immune system, i.e., mitochondrial innate and adaptive immune responses, as well as endogenous antioxidants and DNA repair involving activation of Nrf2 [201]. For example, hormetic antitumor responses were reported by [202]. Mice inoculated with S180 sarcoma cells exposed to 75 mGy whole-body X-irradiation exhibited a significantly attenuated tumor growth with a notable improved erythrocyte immune function as indicated by a decrease of hypoxia factor EPO and VEFGR [202].

Furthermore, epigenetic alterations induced by low-dose IR contributed to radiation-induced hormesis in the mouse (ACvy) [203]. Interestingly, low-dose IR-induced immune hormesis via lncRNAs [204] occurred in an organ-specific fashion [205]. In addition, low-dose IR may function as a hormetin in aging by prolonging the lifetime [206].

It has to be kept in mind; however, that low doses can be beneficial or harmful for human health [207,208].

In a recent detailed review, Scott and Thamalingham summarized present knowledge of low-LET photon-induced hormetic effects mainly involving natural defenses in cells, tissues and whole body and including epigenetically regulated antioxidant production (mitochondria-mediated), DDR with DNA repair selective p53 independent apoptosis of aberrant cells such as transformed and tumor cells, promoting the (mitochondria-dependent) innate and adaptive immune systems with the suppression of inflammatory and the promotion of anticancer immunogenic effects [209]. In particular, non-linearities of low dose (up to 100 mGy) and low-dose-rate effects (1 mGy/day) in comparison to moderate and high doses and dose rates are underlined.

Therefore, in addition to the already mentioned protective effects of IR-induced increases in antioxidants (superoxide dismutase SOD and GSH, glutathione reductase (GR), glutamylcysteine synthetase and Trx GPX [210,211]) low-dose epigenetic changes (DNA methylation, histone modifications, chromatin remodeling and noncoding miRNAs [212]) as adaptive stress responses [213] in cells and developing mice [203], important protective effects of low-dose induced hormesis include stimulation of DNA repair through low-dose (high-background) IR, differences in gene up-regulation at low and high doses and dose rates [214,215], or neoplastic transformation [216], low-dose protective effects on male germ cells [217], reduction of spontaneous lung cancers in mice [218] and protective apoptosis of bystander [219,220], low-dose anti-inflammatory [221] and immunogenic effects [222]. In this context, it should be noticed that low IR doses can suppress cancer metastasis in mice and rats probably via abscopal effects [223,224,225]. Interestingly, no dose-rate effect was observed with high-LET heavy-ion (C-ion) irradiation [226].

In sum, it is difficult fully to grasp LDR-induced hormetic responses. They may be beneficial or damaging depending on the metabolic (mitochondrial) status of the cells (normal vs. cancer cells) and tissues and also on radiation dose, dose rate and quality. The majority of effects are beneficial. They turn on cellular defenses and facilitate cellular activities. However, because of non-linear dose and dose-rate responses, predictions of IR-induced biological impacts at hormetic doses on health risk evaluation remain difficult and need detailed context-specific analyses [227,228]. The whole cellular metabolic and signaling network is involved, as well as mitochondria, at least indirectly. However, case-specific predictive biomarkers are still missing.

#### 3.2.2. Radioadaptive Responses

Genetic and epigenetic mechanisms can impact favorably on radioadaptive responses and the stimulation of cell activities in general. Low IR doses resulting in adaptive responses to challenging high IR doses may be considered to be a preparative measure to provide sufficient energy stocks and to activate the expression of genes involved in cellular defenses (antioxidant, DNA repair, etc.) [194] that are tightly linked mitochondrial activities. The existence of radioadaptive responses involving treatments with a first low priming dose followed a few hours later by a high challenging dose of IR has been ascertained several times since the observations by [229,230]. Interestingly, some reports indicate that low-dose radioadaptive responses involve increased mitochondria-mediated OXPHOS and ATP production in normal cells providing increased energy for DNA repair mechanisms to cope with damage subsequently induced by high-dose IR.

For example, the adaptive response observed in PBMCs from healthy human volunteers after 100 mGy priming dose plus 2 Gy challenging dose of γ-radiation produced a significant increase in cell survival in comparison to a single high-dose exposure (2 Gy) [231]. After the priming dose, the early adaptive response included increased activity of Mn-SOD, catalase, thioredoxin reductase, glutathione peroxidase, increased MMP and metabolism and enhanced translocation of NF-κB and Nrf2 to the nucleus. The results confirmed earlier results obtained by Bravard et al., who showed that a 20 mGy priming dose followed 6 h later by a challenging dose of 3 Gy γ-radiation yielded increases in Mn-SOD (SOD2), catalase and GPX in normal human lymphoblastoid (AHH-1) cells very soon after the priming dose. This clearly showed that mitochondrial functions are involved in the adaptive response [232].

Interestingly, in C3H 10T1/2 mouse embryo cells, the radioadaptive response resulted in a decrease in micronucleus formation and neoplastic transformation frequency [233] and in human cells in a decrease of gene mutations deletion-type mutations [234].

Park et al., showed that the adaptive IR response differed in normal and neoplastic cells [235]. Differences in the low-dose-priming effect were also seen in human fibroblasts and stem cells at the proteomic level [236]. Recently, Wang et al., showed that kV X-rays and MeV X-rays exert different effects in terms of membrane permeability changes, DNA damage induction, cell growth and gene transfer in CHO cells [237]. They observed a threshold for MeV X-ray-induced damaging effects between 1 and 1.5 Gy. In fact, 0.25 Gy promoted cell growth, whereas 1.5 Gy induced cell damage. Very low doses of 120 kV X-rays (0.01–0.04 Gy) induced no obvious cell damage.

Jiang et al., reported that low-dose IR-induced adaptive responses occurred in normal cells but not in tumor cells [238]. This finding was later confirmed by Yu et al. [239].

Concerning radiation quality effects, Varès et al., extended previous studies in showing that after a low-LET priming dose of 0.02–0.1 Gy of X-rays followed 6 h later by a challenging dose of 1 Gy by high-LET heavy-ion irradiation (C-ions, 20 and 40 keV/μm) or neon ions (150 keV/μm) yielded an adaptive response on mutations induced in human lymphoblastoid cells, probably due to modulation of DNA DSB repair capacity for complex DSBs and clustered DNA damage [240].

In addition, teratogenic effects induced in fetal mice could be attenuated by an adaptive response using an X-ray priming dose followed by exposure to monoenergetic C-ions, neon ions, silicon (LET values 15, 30, 55 keV/μm) but not when using iron ions (200 keV/μm) for the challenge dose [241,242]. This suggests possible specific radiation quality effects of the challenge radiation.

In addition, radioadaptive responses were observed regarding IR-induced tumorigenicity. Moreover, Zhou et al., indicated an interesting pivotal role of the immune system in low-dose IR tumor inhibition in Lewes lung-cancer-bearing mice [243]. Low doses followed by a high-dose IR (1 Gy) resulted in lower tumor growth than with high-dose IR due to the activation of the adaptive immune system with the activation of T cells and NK cells.

From a mechanistic point of view, it can be concluded that DDR and DNA repair pathways (p53, ATM and PARP), antioxidant response (Nrf2 pathway), cell survival/apoptotic death pathways, endoplasmic stress response (UPR), immune/inflammatory response (NF-κB pathway autophagy (mitophagy)) and cell cycle regulation (cyclin B1/CDK1 complex) are implicated in adaptive responses [244] as well as the translational machinery [245].

#### 3.2.3. Low-Dose Radiation Hypersensitivity (HRS)

Low-dose IR-induced hypersensitivity is characterized by high initial clonogenic radiosensitivity of mammalian cells observed at low doses ranging from <0.1 Gy to 0.3 Gy followed by radioresistance at higher doses [246,247,248]. To explain this phenomenon on hyperradiosensitivity (HRS), several hypotheses have been advanced:(1)Ineffective cell cycle arrest in the G2/M phase [248,249] and links to early G2/M checkpoint activation [250,251].(2)A putative damage-sensing threshold for triggering faster or more efficient DNA repair (threshold around 0.3 Gy) [246,252]) was substantiated by Fernet et al [253].(3)Links to bystander and NTE [254,255,256,257,258].(4)Mitochondrial metabolism involving ATP [259].(5)Involvement of ATM in HRS [254,260,261].(6)Change in the balance between the induction of HRS and subsequently induced radioresistance (IRR) depending on cell type.

Only a few data evoke the possible contribution of mitochondria-mediated adaptive effects on HRS. The phenomenon of low-dose radiosensitivity involves very limited reactivity of cell populations in the G2/M phase, which are employing most of their available energy (ATP) to ensure mitosis and cell proliferation [259]. In G2/M, stalled DNA replication forks may increase mitochondrial ROS and limit ATM/ATR activation leading to cell death in a fraction of the cell population. In fact, the first dip in cell survival curves at low initial doses <0.3 Gy may be caused by a similar phenomenon: cells in G2 that are going to provide energy are short in energy because of insufficient energy reserves in the cell. During progressively higher doses, cells may be able to activate mitochondrial energy production. This may involve also the late oxidative activation of ATM in the G2/M phase [261,262]. As shown by Enns et al., DNA DSB repair is more efficient during IR-IRR than during HRS. Interestingly, in normal cells, ATM is not phosphorylated following 0.1 Gy γ-irradiation up to 4 h post-irradiation but is increased up to four-fold after 0.25 Gy with increased DSB induction (as revealed by γH2AX) [262]. However, the reduction in DSBs after 24 h although DNA repair was more effective after 24 h post-irradiation than after exposure to 0.1 Gy. The authors confirmed that ataxia-telangiectasia cells (defective in ATM) did not show IRR [262]. This is interesting because ATM is also involved in the management of late oxidative stress including mitochondrial oxidative stress induced by IR in the G2/M phase [126] as well as in bystander effects in cooperation with ATR [263].

Heuskin et al., reviewed the induction of low-dose hypersensitivity by X-rays, protons, α-particles and C-ion irradiation in V79 and human cells [264]. Reportedly, results were similar for sparsely ionizing and densely ionizing radiation except at very high LET. Mechanistically, the data indicated that insufficient ATM induction in G2/M cells and the absence of IR-induced cell cycle arrests in G2/M were involved in HRS. In addition, Cherubini et al., analyzed hypersensitivity, IR-IRR, adaptive response and bystander effect in V79 Chinese hamster cells and human glioblastoma cells using protons and C-ion irradiation [255].

Exposure to heavy ions such as C-ion irradiation (75 keV/μm) was shown to induce HRS in normal human fibroblasts [260,261], involving the cooperation of ATM with ATR. Xue et al., and Ye et al., noticed that intercellular gap junctions were involved in NTE and bystander effects and in C-ion-induced HRS [254,261]. Using X-ray microbeam irradiation, Maeda et al., showed that exposure of the nucleus enhanced low-dose IR-induced HRS, whereas cytoplasmic exposure involving mitochondria suppressed HRS, and bystander effects were associated with HRS effects [265,266]. After X-irradiation, ATM-mediated DNA repair and transfer of healthy mitochondria to neighboring cells via microtubule ATM-dependent exchange appears to be involved in HRS as well [267]. Co-cultures of X-irradiated ATM-deficient cells were unable to exert mitochondrial transfer to neighboring cells. This clearly confirms that mitochondrial low-dose IR responses are implicated in HRS.

### 3.3. High-Dose IR Responses and Signaling Involving Mitochondria

While intermediate IR doses (0.1–1 Gy) start to exhibit significant effects on survival and cell proliferation, high doses (i.e., radiotherapeutic doses >1 Gy) are usually associated with severe damage, cell cycle arrest, cell death (apoptosis), inflammatory and immunogenic responses of murine and human cell systems. These effects are reflected by transcription of the specific gene families induced [181,182,215]. They also rely on epigenetic control mechanisms and involve mitochondrial functions. Even if these are not yet fully understood, they may well have a bearing on radioprotection and radiotherapeutic issues.

#### 3.3.1. IR-Induced DNA Damage and DNA Damage Response (DDR) Signaling

As indicated above, responses of mammalian cells to the induction of IR damage in genomic DNA, mtDNA and mitochondria involve a large network of interactions. Multiple types of radiation damage need to be recognized and handled by several pathways to ensure cell survival. Cell death is induced when damage is excessive [4].

In nDNA, the most deleterious lesions, DNA DSBs and clustered lesions can be induced by low-dose LET IR, but are mostly induced by high-LET IR and heavy ions. For example, C-ion irradiation gives rise to higher levels of DNA DSBs, complex clustered DNA [268,269] and lethal damage [265,270,271,272,273,274,275] than low-LET photon IR. The repair of complex clustered lesions needs the interaction of components of different repair systems [275,276,277,278,279,280]. High-LET IR (C-ions) is characterized by incomplete DNA repair [269], long-lasting cell cycle arrest and high rates of apoptosis [261,281,282,283,284].

IR-induced DDR highly depends on mitochondrial metabolic functions and mitochondrial signaling. For example, incomplete DNA repair and accumulation of unrepaired or misrepaired lesions initiate intracellular signaling toward involving mitochondria-mediated signaling, bystander effects and immunogenic and/or immunosuppressive responses leading to programmed cell death (apoptosis) and antitumor responses (see Figure 1).

However, low- and high-LET irradiation elicit quite different DDRs: for instance, in lung adenocarcinoma A549 cells, C-ions (LET 290 keV/μm) at 1 Gy induced ATM/ATR activation more strongly than γ-rays at 4 h post-irradiation [285]. In addition, foci of repair clusters (e.g., BRCA1 foci) could be detected after C-ions which were larger than those induced by low-LET γ-rays suggesting a specific response to C-ion-induced clustered DNA damage. Furthermore, the level of the ATR-activated checkpoint protein Chk1 (phosphorylated Chk1) was much higher after C-ion than after γ-ray exposure, but Chk2 phosphorylation was observed in both irradiation conditions. Compared with low-LET X-rays (320 keV), Dokic et al., found relative biological effectiveness (RBE) values for C-ion exposure (100 keV/μm) were 2.56 and 2.20 in two glioblastoma cell lines (NCH82 (radiosensitive), NCH149 (radioresistant)), respectively [286]. Radioresistant cells (NCH149) showing lack of ATM signaling and absence of ATM phosphorylation associated with intrinsic chromosomal instability were clearly more sensitive to C-ions (100 keV/μm) than to X-rays [286].

Recently, it has been hypothesized that after high-LET particle exposure, the ionization tracks may not be sufficiently dispersed to interact with ATM dimers causing ATM activation and the DDR response. As a consequence, recognition of DSBs would be impaired [287]. The authors indicated a possible link to the chromosomal instability status of cells and the possible absence of p53 and CHK2 activation (phosphorylation) but did not check effects on mitochondria. Interestingly, Wozny et al., showed that under hypoxia, silencing of HIF-1α that is translocating to the nucleus significantly radiosensitizes HNSCC and cancer stem cell subpopulations to photons and C-ion exposures because of decreased DSB detection by the ATM protein and decreased DSB repair by HR and NHEJ [288].

High-LET IR (C-ions) usually causes stronger and more persistent cell cycle arrests, leaving more DNA damage less accurately or unrepaired, and thus promoting more strongly subsequent programmed cell death pathways than low-LET IR.

In general, the DDR response constitutes a fairly early response, whereas mitochondria-mediated signaling and apoptotic cell death can be relatively late and may include long-term inflammatory and immunogenic responses.

#### 3.3.2. Involvement of IR-Induced ROS Production in G2/M Cell Cycle Arrests, MN Formation and Persistent Oxidative Stress

Yamamori et al., clearly showed that increased mitochondrial ROS production by IR (200 kVp X-rays, at 10 Gy) in human lung cancer cells A549 was accompanied by an up-regulation of mitochondrial ETC function and cell mitochondrial content, leading particularly to cell cycle arrest in the G2/M phase [177]. The X-ray-induced G2/M cell cycle arrest at 24 h post-irradiation increased mitochondrial respiratory complex II and III activity, the amount of mtDNA, overall mitochondrial ROS levels and oxidative stress. Likewise, a high single dose (>5 Gy) of X-rays induced G2/M cell cycle arrest, increased mitochondrial mass mtDNA and ROS in normal human neural stem cells (NSCs) without increasing mitochondrial activity [125].

There is also evidence that mitochondria-dependent antioxidant levels (GSH) influence the proportion of cells arrested at the G2/M checkpoint in culture depending on radiation quality (10–15% of cellular GSH is located in mitochondria to protect against ROS generated during coupled electron transport and OXPHOS [289]). At biologically equivalent doses of C-ion irradiation (33.4 keV/μm), cell cycle arrests in G2/M were significantly prolonged, mitochondrial functions altered, ROS production increased and DNA repair slowed in comparison with X-irradiation in a head-and-neck radioresistant cell line SQ20B [290].

The higher efficacy of high-LET particle exposures in cell cycle arrest was proven several times in human cancer cells: by Suetens et al., with carbon ions (^13^C-ions, 33.7 keV/μm) in colorectal adenocarcinoma (Caco-2) and melanoma cells (HTB140) [291], Hartfiel et al., with ^12^C-ions (101 keV/μm) in adenoma carcinoma and esophageal cancer cells [269], by Zhang et al., with ^12^C-ions (50 keV/μm) in breast cancer cells (MDA-MB-231 and MCF-7) [292], by Habermehl et al., with ^12^C-ions (112 keV/μm) in hepatocellular carcinoma Hep3B cells [293] and Matsui et al., with ^12^C-ions (87 keV/μm) in cervical cancer HeLa cells [294]. Moreover, Matsui et al., showed that the human oxidation resistance 1 gene (OXR1) played a more important role after the C-ion-beam than after ^56^Fe-ion-beam (200 keV/μm) irradiation [294].

ATM is involved in the regulation of cellular ROS levels, mitophagy, protein homeostasis and ROS-dependent autophagy and in the activation of cell cycle checkpoints [295]. Interestingly, ATM/ATR was induced more strongly by γ-rays at 1 Gy than by C-ions (LET 290 keV/μm) at 4 h post-irradiation in lung adenocarcinoma A549 cells [285]. Moreover, late ATM induction due to oxidative stress production was found to be absent after high-LET C-ion irradiation [169].

#### 3.3.3. Mitochondria-Mediated Programmed Cell Death

Most often, metabolic stress or toxic insults from various chemical and physical agents (including IR) lead to cell cycle arrests and programmed cell death (apoptosis). Several excellent reviews are available [296,297,298]. Mitochondria are prominent players in several regulated cell death pathways. Cell death can be directly induced by the damage inflicted on DNA or indirectly by oxidative radicals as a consequence of IR exposure and ROS generated by mitochondria and metabolic dysfunction.

The best-known cell death pathways are those following the induction of severe damage to nDNA involving activation of DDR signaling and initiation of mitochondria-mediated programmed cell death (apoptosis). Less is known about the role of mtDNA damage in IR-induced cell death. Specific mutations such as the “common deletion” induced in mtDNA (low 0.1 Gy and therapeutic >1 Gy doses) could lead to cell death via the induction of mitochondrial dysfunction and the generation of ROS promoting mitochondria-mediated intrinsic or extrinsic apoptotic pathways [299]. Moreover, mtDNA released from IR-damaged mitochondria can be detected using the cGAS-STING pathway and provoke an immunogenic type of cell death [300] (see the section on immunological IR effects below).

Among the forms of regulated cell deaths, one can distinguish intrinsic apoptosis (mitochondria-driven, breakdown of mitochondria, condensation of nuclei, DNA cleavage apoptotic bodies), extrinsic apoptosis (membrane receptor-driven by death-inducing signaling complex involving also mitochondrial components such as MOMPs) and others such as mitochondrial membrane permeability transition (mitochondrial-driven transition pore (MTP)-necrosis (mitochondrial swelling and MOM rupture), ferroptosis (mitochondrial shrinkage, rupture, cristae disorder involving ROS), pyroptosis (condensation of chromatin, plasma membrane, permeabilization), parthanatos (PARP activation, DNA fragmentation, mitochondrial permeability, apoptosis-inducing factor (AIF) activation), entotic cell death (engulfment by non-phagocytic cells observed in neoplasia)), netotic cell death (membrane degradation involving NADPH oxidase-mediated ROS production, autophagy and release and translocation of granular enzymes from the cytosol to the nucleus), necrosis (cell swelling and rupture), necroptosis (necrotic membrane rupture, DAMP release), lysosome-dependent cell death (autophagy), autophagy-dependent cell death (autophagic cell death), alkaliptosis and oxytosis/oxeiptosis (necrotic mitochondrial shrinkage, rupture, cristae damage) and cellular senescence, mitotic catastrophe (failed mitosis, including giant cell formation, polyploidization and anaphase bridges) and immunogenic cell death [296,297,301].

Ferroptosis is involved in p53-mediated radiosensitization in RT via GSH depletion and promotion of lipid peroxidation [302,303,304]. However, the involvement of mitochondria in IR-induced ferroptosis is not yet proven [305,306]. Apoptosis, necroptosis and ferroptosis may be associated with selective autophagy (mitophagy) [303].

Mitotic catastrophe, anoïkis and pyroptosis refer to mechanisms executed by the apoptosis and necrosis machinery. Anoïkis results from the lack of cellular attachment to the extracellular matrix and involves mitochondrial functions of the intrinsic and extrinsic apoptotic pathways [306]. Inflammatory pyroptosis involves pattern receptor signaling, inflammasomes NLRP3 and NLRC4, caspase-1 ctivation, activation of gasdermin, plasma membrane pore formation and activation of the innate immune system via release of cytokines IL-1β and IL-18 [301]. Pyroptosis has some features in common with apoptosis [307] but is thought to be responsible for unwanted side effects and tissue toxicity observed after conventional RT [308].

Thus, mitochondria can be (directly or indirectly) involved in several death pathways associated with oxidative stress, such as oxeiptosis (anti-inflammatory), parthanatos (pro-inflammatory), necroptosis (pro-inflammatory), possibly ferroptosis (pro-inflammatory), autophagy-dependent and lysosome-dependent cell death (inflammatory), anoïkis (resistance to metastasis)-dependent cell death due to inhibition of autophagy or mitophagy [297].

Here, we focus on the two main mitochondrial pathways known to operate after IR exposure, the mitochondria-mediated intrinsic and extrinsic pathways [298] as well as a minor pathway, the ceramide-dependent pathway. These pathways lead to the activation of caspase-3 and -7 and subsequent cell degradation [298] (Figure 2). Mitochondria are needed during initiation but also for metabolic energy supply during all execution phases [309].

##### Intrinsic Mitochondrial Apoptotic Pathway

Following IR exposure, the intrinsic mitochondrial apoptotic pathway in p53 competent (wt) cells starts with activation of DDR, p53 and balanced activation of proapoptotic Bcl-2-associated X protein (Bax) and antiapoptotic (Bcl-2) factors, e.g., the Bcl-2 antagonist killer (Bak), mitochondrial outer membrane permeabilization (MOMP), reduction in MMP, and proceeds with the release of cytochrome c from mitochondria and activation of the caspase-9 pathway leading to cell death. As indicated by Golden et al., after the induction of DNA damage in p53 wildtype cells, ATM becomes activated and phosphorylates p53 on serine 15 [310]. This leads to activation of PUMA and some other proteins, which translocate to the cytoplasm followed by disruption of the Bcl/XL/p53 complex and the liberation of p53. In turn, the disruption of the complex Bcl-2/Bax by free p53 results in the release of proapoptotic Bax. The latter permeabilizes the outer mitochondrial membrane (MOMP) and triggers the release of cytochrome c from the mitochondria into the cytoplasm. An apoptosome (cytochrome c, Apaf-1 and ATP) is formed and caspase-9 activated, which initiates the action of caspase-3, -6 and -7 and subsequent cell degradation including DNA fragmentation. This also involves the release of AIF [311].

##### Extrinsic Mitochondrial Apoptotic Pathway

The extrinsic pathway depends on external damage signaling through binding of tumor necrosis factor (TNF) family ligands to death receptors of the plasma membrane, i.e., Fas ligand 1 (CD95L), Faa5 (CD95) and FAAD leading to the activation of extrinsic pathway-specific pro-caspase-8, activation of caspase-8 and subsequent activation of caspase-3, -6 and -7 and cell death [298,312].

For example, increased death receptor Fas (CD95) expression in extrinsic apoptosis could be induced by low-LET IR in lymphocytes of C57/B6 mice [313], in rat C6 glioma cells [314], in human p53 wt breast cancer cells (MCF-7) [315], in human nasopharyngeal carcinoma cells [316] and in HeLa cells [317]. The latter required c-Jun N-terminal kinase (JNK) for up-regulation of Fas and subsequent activation of Bax and Bak and the mitochondrial apoptotic cell death pathway [317].

The intrinsic and extrinsic pathways converge at early MOMP [311]. In the extrinsic pathway, this constitutes one of the early steps leading to the activation of the initiator protease caspase-8, its substrate BH3-interacting-associated death antagonist (Bid) and the downstream effector Bax and apoptotic degradation [318]. Mitochondrial membrane permeabilization and cytochrome c release may be further increased by the destabilization of the MOM by externalized oxidized cardiolipin.

##### The Ceramide-Dependent Pathway

As shown by Ardail et al., γ-ray exposure of human Jurkat leukemia E6.1 cells triggered sphingomyelin hydrolysis with *de novo* synthesis of ceramide and a ceramide response in plasma membrane rafts leading to acid sphingomyelinase targeting of mitochondria and mitochondria-mediated apoptosis [319]. In addition, Sia et al., reported that after exposure to IR, a ceramide-dependent pathway can be initiated as well involving IR-induced activation of acid sphingomyelinase in the plasma membrane, production of ceramide through hydrolysis of sphingomyelin [298]. Ceramide acts as a second messenger for the initiation of apoptosis. This pathway (Figure 2) feeds then into the final steps of the mitochondrial intrinsic apoptotic pathway with activation of caspase-3 and -7 and subsequent cell degradation [298]. Ceramide is in the center of sphingolipid metabolism and is part of mitochondrial lipid signaling [318]. Exposure to IR leads to sphingolipid accumulation on the plasma membrane [320]. In the initial phase, it is processed into two lipid messengers (sphingosine 1 phosphate and hexadecenal), which activate Bax and the Bcl-2 antagonist killer, BaK, on the mitochondrial membrane. Ceramide can induce apoptosis by triggering Bax-dependent MOMP and targeting of the voltage-dependent anion channel 2 (VDAC2) as also found after cardiolipin extrusion [318], which constitutes an “eat me” signal in mitophagy. Moreover, an interesting crosstalk exists between cardiolipin and the ceramide-directed pathway. Cardiolipin can inhibit ceramide synthesis and mediate its cleavage by ceramidase and apoptosis (see [318]). Interestingly, Alphonse et al., showed that the γ-ray induction of ceramide in the mitochondrial caspase cell death pathway was defective in radioresistant SQ20B head and neck tumor cells but present in radiosensitive Jurkat and moderately sensitive SCC61 tumor cells [321]. Radiosensitization was associated with a decrease in MMP, increased ROS, decreased glutathione levels and activation of caspases. Independently of p53 status, C-ion exposure (33.4 or 184 keV/μm) induced early ceramide production in the radiosensitive and late production in the radioresistant cell line correlated with early and late apoptosis, respectively [322]. In radioresistant cells (SQ20B), after late apoptosis, mitotic catastrophe followed.

Recently, Ferranti et al., confirmed activation of acid sphingomyelinase, generation of ceramide at the plasma membrane a few minutes after γ-irradiation and a dose and time-dependent induction of apoptosis in hematopoietic Jurkat cells [323]. Electron microscopic analysis showed that the first step after IR exposure (10 Gy) consisted of the formation of nanopore-like structures in the plasma membrane due to lipid peroxidation leading to calcium entry, initiation of lysosome fusion and subsequent apoptosis (40–50% at 24 h post-irradiation).

#### 3.3.4. Autophagy

Autophagy is a general autoprotective cellular process in which cytoplasmic cargoes are taken up by double-membrane vesicles, i.e., autophagosomes, engulfed by lysosomes and degraded [324]. For example, dysfunctional mitochondria can be eliminated by a selective type of autophagy called mitophagy if mitochondria cannot be restored by fusion and fission or otherwise. Defective proteins, membranes, the ER, peroxisomes, bacteria and virus particles are subject to selective autophagy as well [325].

Autophagy involves membrane distention from the ER or Golgi complex and formation of the pre-autophagosome followed by marking damaged organelles (e.g., dysfunctional damaged mitochondria) by ATG proteins and LC3 and enclosure by the autophagosome. After fusion with lysosomes, final degradation occurs [324].

As noted by Jin et al., autophagy can also start with the accumulation of PINK1 on the outer membrane of depolarized or damaged mitochondria followed by the recruitment of PARK2/Parkin, an E3 ubiquitin ligase [326]. Parkin 2 mediates the ubiquitination of several outer membrane proteins including MFN1 and MFN2. The sequestosome SQSTM1 (p62) can bridge the ligand on the MOM with the autophagy machinery. Other proteins may also be involved.

How exactly exogenous signals from IR and chemotherapeutics, or endogenous signals from redox signaling due to hypoxia, link the generation of ROS and mitochondrial fragmentation and fission to apoptotic responses is not yet fully understood. After IR exposure, external or internal attacks by ROS may open the permeability transition pores of inner mitochondrial membranes leading to loss of membrane potential and loss of import of cytosolic proteins. The damaged and partially dysfunctional mitochondria may then undergo mitoptosis and subsequent removal by selective autophagy (mitophagy) to either retain healthy and fully functional mitochondria, or, when excessively damaged, they may undergo apoptosis [41].

#### 3.3.5. Mitophagy

Selective autophagy, i.e., mitophagy, is part of the mitochondrial quality control system that eliminates damaged and dysfunctional mitochondria and fuels remaining healthy mitochondria to fission and fuse to maintain and restore metabolic homeostasis in normal cells. Basically, mitophagy serves to ensure normal cellular metabolic functions and adjust the number of mitochondria per cell to energetic needs [327,328,329,330,331,332]. For example, mitophagy eliminates mitochondria from developing erythrocytes and contributes to the maternal inheritance of mtDNA through the elimination of sperm-derived mitochondria [327].

Because of their high dependence on intact mitochondrial functions, tissues such as the heart, brain, liver, kidney, and skeletal muscle exhibit increased basal levels of mitophagy [332]. However, this process can also fuel cancer cells with new energy for cell survival, proliferation, evasion, migration and metastasis and can play an important role in several human diseases [333].

Apparently, mitophagy can be elicited in different ways. Mitophagy may be initiated by a disruption of MMP as an important trigger [334]. The general scheme is that dysfunctional mitochondria are recognized by the autophagosomes which are taken up by lysosomes for final degradation [47,324,326]. Mitophagy can be initiated by mitochondrial ROS due to OXPHOS or mitochondrial dysfunction following excessive oxidative stress, mutations in mtDNA or damage to proteins [47]. It proceeds as follows: mitochondrial substrates are first ubiquitinated on the mitochondria and become recognized by LC3 adapters such as the sequestosome-1 (SQSTM1) p62, optineurin (OPTIN) and others. OPTIN serves to recruit the autophagosome to the mitochondria. The process may include the protein Parkin, which is involved in the initiation step of autophagy and the clearance of dysfunctional mitochondria by mitophagy [335]. Impaired mitochondria with low MMPs are recognized by Parkin. In fact, DNA damage (oxidative damage and DNA DSBs) was shown to provoke translocation of the E3 ubiquitin ligase Parkin to the nucleus and promotion of DNA repair activities in human cancer cells (HeLa, SH-SY5Y neuroblastoma) [336]. As recently shown by Correia-Melo et al., ectopic expression of the ubiquitin E3 ligase Parkin combined with short-term uncoupling treatments can stimulate widespread mitophagy and effective elimination of mitochondria [337].

Besides being involved in DDR, the protein ATM may also operate in sensing oxidative stress in the cytosol. It can modulate the Pink1–Parkin pathway and promote the activation of autophagy (by inhibition of the regulator mTOR complex 1 (mTORC1)), mitophagy elimination of altered mitochondria by mitophagy, and pexophagy (i.e., selective peroxisome removal) [338]. However, in B-cell lymphoma cells, mitophagy appeared to be independent of ATM kinase-mediated phosphorylation of Parkin (a regulator of mitophagy), Pink1 and Parkin-Ub (Ser65) [339].

Moreover, in mammalian cells, mitophagy can be receptor-mediated, via the NIP-3-like protein 3 (NIX) located on the MOM [330] favoring mitophagy [340]. Accumulation of ROS promotes NIX-mediated mitophagy and the recruitment of LCR3 to mitochondria [29,30,341]. Interestingly, IR-induced mitophagy may be independent of conventional autophagy involving the accumulation of lysosome-like organelles in mitochondria through Spata18 [342].

During mitophagy, cardiolipin, a membrane lipid located in the inner membrane of mitochondria is also translocated to the outer membrane (externalization) following toxic insults, e.g., IR and rotenone [47]. There, it can act as a signal for subsequent degradation with the help of autophagy protein microtubule-associated 1-light chain-3 (LC3). LC3 mediates autophagosome and cargo recognition and engulfment of mitochondria by the autophagic system (e.g., mitophagy) [47]. In addition, externalized cardiolipin serves as an “eat me” signal for macrophages in immunological responses [343], especially in neuronal cells.

In damaged mitochondria, PINK1 (a mitochondrial serine/threonine-protein kinase) may also accumulate to the MOM following depolarization of the mitochondrial membrane. After activation by autophosphorylation of PINK 1, subsequent phosphorylation of Parkin 1 and polyubiquitination of MFN1/2 and other substrates (e.g., VDDAC1 and Miro1) may then result in the degradation by the proteasome. In this process, degradation of MFN1/2 is associated with the induction of mitochondrial fission and also with mitophagy [344].

Interestingly, in ATM and NBS complemented cell lines and normal human fibroblasts, Shimura et al., observed an increase in mitochondrial mass and mitochondrial biogenesis together with accumulation of mitochondrial ROS after very-low-dose fractionated low-LET IR exposure (150 kVp X-rays for 31 days 10 mGy or 50 mGy fractions, total dose 0.46 or 2.3 Gy 46) [345]. The mitochondrial damage arising from excess ROS was recognized by Parkin-initiated mitophagy. Irradiated ATM- and NBS-deficient cells turned out to be defective in mitophagy, accumulated abnormal mitochondria showing fragmentation and a decrease of MMP. This led to the accumulation of damaged mitochondria, release of cytochrome c, activation of caspases and apoptosis.

In addition, high-LET C-ion beam exposure could activate Parkin in HeLa cells leading to extensive caspase-dependent apoptosis. Moderately damaged mitochondria by low doses of C-ion irradiation (0.5 Gy), were eliminated by complete mitophagy [346]. The removal of dysfunctional obsolete mitochondria via mitophagy could inhibit the leakage of ROS and possibly the release of mtDNA to cell surroundings and over the whole body [324].

In any case, mitophagy is part of strict quality control. It reduces cellular oxidative stress due to ROS and helps to maintain stable mitochondrial functions and energy supply [347].

For the maintenance of mitochondrial homeostasis, mitochondria are also in contact and interaction with other organelles such as ER, Golgi apparatus and lysosomes [21]. This also includes the so-called UPR, which consists of intramitochondrial control of unfolded proteins involving ER as well as extramitochondrial control by mitophagy. Correctly folded proteins involved in mtDNA repair (such as mtDNA polymerase γ) are imported by chaperones. However, damaged proteins of the MOM may undergo ubiquitination by MARCH5, extraction by p97 and degradation by the proteasome. Specific mitochondrial proteases in the mitochondrial inner membrane space (mitochondrial matrix) degrade altered proteins, whereas damaged, dysfunctional mitochondria are eliminated by mitophagy [47].

The exact pathway of mitophagy differs depending on the type of the initial cellular insult and the mitochondrial stress encountered [332].

Mitochondrial quality control mechanisms (autophagy and mitophagy) are of particular importance for cancer cell dynamics following exposures to damaging agents such as IR (see the section on IR-induced autophagy and mitophagy below). In cancer cells, shifts from normoxia to hypoxia can be accompanied by increased mitophagy to adapt cells to the change in energy supply under anaerobic conditions [348]. Mitophagy is of great importance in anticancer and anti-metastatic radiotherapy (RT), immune responses [349,350] but also for radioprotection of the heart [351]. Very importantly, tissues with high energy (ATP) consumption, i.e., brain, muscles, heart, liver, kidney as well as cancer cells and tissues exhibit high numbers of mitochondria [29,30] the function of which can be strongly affected by IR-induced damage.

### 3.4. Impacts of Low- and High-LET IR (Heavy Ions and C-ions) on the Induction of Apoptosis, Autophagy/Mitophagy and Anastasis

The apoptotic responses of normal and cancer cells to low-LET IR (photons, X-rays and γ-rays) and high-LET ion irradiation (particle radiation, α-rays and heavy ions) appear to differ. Low-LET IR usually induces only slight mitochondria dysfunction, fusion and fission, whereas high-LET IR such as heavy ions (C-ions) induces significant increases in mitochondrial dysfunction, fission, mtDNA fragmentation and reduced fusion. These radiation quality effects determine molecular pathways involved in mitochondrial-mediated apoptosis and the effectiveness of irradiation exposures. IR initiates apoptosis mostly via the induction of the DDR and the enhancement of the free-radical-driven generation of ROS, which can be especially exacerbated by exposure to heavy ions (C-ions). DDR involves activation of p53, cell cycle arrests followed by either error-free or error-prone DNA repair or apoptosis. Thus, IR exposure can lead either to p53 mutations and radioresistance or, in the presence of excessive damage, to apoptosis [352].

In p53 wt cancer cells, DDR-dependent early apoptosis can be induced by both low- and high-LET IR. High-LET IR shows a higher efficacy in apoptosis induction on radioresistant cells [353] than low-LET [354]. Interestingly, induction of apoptosis was found to be especially effective by high-LET heavy ions such as C-ions and independent of cellular p53 status [322,355,356,357,358,359,360].

As shown in Table 1, there is ample evidence that low-LET irradiation (X-rays and γ-rays) are significantly less efficient in inducing apoptosis than exposure to heavy ions taking C-ion exposure as an example [285,291,293,322,324,346,353,356,360,361,362].

This applies to normal cells and even more to cancer cells. IR-induced apoptosis usually follows an efficient block in the G2/M phase, which is clearly more stringent after C-ion than after X-ray exposures. For example, C-ions (197 keV/μm) induced a strong G2/M cell cycle block in radioresistant human HTB140 melanoma cells and, dose-dependently (2–16 Gy), mitochondria-mediated apoptosis (up to 21.2%) [370]. In human breast cancer cell lines MDA-MB-231(radioresistant cancer stem cells) and MCF-7, C-ions of 50 keV/μm (at 4 and 8 Gy) were clearly more efficient in inducing G2/M cell cycle arrest and apoptosis than X-rays [292]. In line with this, heavy-ion beams (^12^C-ions (101 keV/μm) and ^16^O-ions (141 keV/μm) had a greater inhibitory effect on cell proliferation and cell cycle progression in the G2/M phase of malignant melanoma cells than low-LET photons [362].

As already outlined above, IR-induced programmed cell death (apoptosis) follows the mitochondria-mediated intrinsic and/or extrinsic apoptotic pathway in most cases.

The involvement of mitochondria has been convincingly shown by [346,369]. Induction of apoptosis in human cervical cancer cells (HeLa) was lower with 3 Gy X-rays than with C-ions (70 keV/μm) [369]. High-dose C-ion exposure significantly increased the expression of the DRp1, a GTPase localized in the cytoplasm but recruited to mitochondrial sites of fission via interaction with the mitochondrial FIS1 but considerably decreased the mitochondrial outer membrane GTPases mitofusin 1 (MNF1) and mitofusin 2 (MNF2) as well as the inner membrane GTPase optic atrophy 1 protein (OPA1) regulating fusion of the inner mitochondrial membrane. In human breast cancer cells (MDA-MB-231), Jin et al., observed at 0.5 Gy of C-ion irradiation (75 keV/μm) a moderate fragmentation of mitochondria, which was considerably increased at 3 Gy associated with significant increases in Drp1 and decreases in MNF1, MNF2 (fission) and OPA1 (fusion) [346]. Interestingly, at a low dose (0.5 Gy) of C-ions, depletion of Drp1 or Fis1 suppressed fission and cytoprotective mitophagy but enhanced apoptosis and drastically reduced survival. The authors concluded that in cancer cells, low-dose C-ion exposure (0.5 Gy) may induce mitochondrial fragmentation but ensure mitophagy and cell survival, whereas high-dose C-ion (3 Gy) exposure would induce serious mitochondrial fragmentation and strong apoptotic cell death (up to nearly 60%) in cancer cells. This demonstrates that there is a close link of high-dose C-ion irradiation-induced mitochondrial fragmentation to high-rate apoptosis.

In general, high-LET irradiation induces stronger apoptosis than low-LET irradiation (Table 1). As indicated by Held et al., the RBE for mitochondria intrinsic apoptosis with caspase-9 and caspase-3 activation is LET-dependent and may culminate at RBE values of 3–4 for helium, carbon, neon and argon ions for a LET of about 100 keV/μm and for boron, silicon and iron ions above a LET of 100 keV/μm [371]. ^56^Fe ions (200 keV/μm) were about eight times more effective than X-rays [364].

Very importantly, DDR and mitochondrial-mediated apoptosis are interconnected, and IR affects all the different steps of apoptosis (Figure 3). In most cases, high-LET irradiation by heavy ions such as C-ions affects the different steps much more strongly than low-LET irradiation, or at least with equal efficiency.

The predominant intrinsic pathway is thought to involve Bax and Bak in the initiation phase and the release of cytochrome c, Apaf-1 and AIF leading to caspase-9 activation and subsequent caspase-3 activation [372], whereas the extrinsic pathway involves death receptors CD95/APO-1/Fas, TNF-β and TRAIL to trigger caspase-8 [373].

The initiation phase involves induction of damage by IR, induction of DDR, mitochondrial damage and oxidative stress. This leads to (1) MOM changes in MMP (ΔΨ_m_), (2) IR-induced activation of Bax and Bak and the release of cytochrome c, (3) IR-induced MOMP, (4) IR-induced effects on ATP production, (5) mitochondrial fragmentation, (6) IR induction of caspase activity and PARP cleavage, (7) involvement of second mitochondria-derived activator of caspase (SMAC) in IR-induced apoptosis and (8) IR-induced release of AIF, endoG and CAD.

(1)IR-induced decrease in mitochondrial membrane potential (MMP)

In Jurkat cells (radiosensitive) and SCC61 (moderately radiosensitive) cells after 10 Gy γ-irradiation with an initial high MMP at 8 h post-irradiation, Alphonse et al., observed a decrease of MMP that preceded a strong increase of cellular ROS [321]. Following X-irradiation of human lung cancer A549 [177] and HL60 leukemic cells [374] a close link was observed between ROS induction, change in membrane potential and induction of late apoptosis. In contrast, in hepatic cancer HepG2 cells, heavy-ion (C-ion (350 MeV/u)) irradiation (4 Gy) induced ROS, decreased the MMP and significantly increased mtDNA damage (8-OHdGua), activation of NADPH oxidase and cell death (apoptosis) within a few minutes [375]. Exposures to carbon ions (350 MeV/n (LET: 50 keV/μm)) and protons (3 MeV, LET: 10 keV/μm) by spot application on mitochondria resulted in a loss of MMP [376]. C-ions were much more effective than protons because 100 carbon ions or 3500 protons were needed for total mitochondrial membrane depolarization. In addition, a dose of 2 Gy of C-ion (30 keV/μm) exposure to human lung cancer cells NSCLC (A549 and H1299) caused a significant decrease in MMP, reduction in ATP production and increase in Ca^2+^ signaling and apoptosis [377]. The data suggest that high-LET C-ion exposures cause a strong decrease of MMP in cancer cells.

(2)IR-induced activation of Bax and Bak and the release of cytochrome c

C-ions in mice up-regulated expression of pro-apoptosis factors, Bax and cytochrome c, and down-regulated expression of apoptosis-profilin (Bcl-2, survivin) and proliferation-related proteins (proliferating cell nuclear antigen). The results indicated that radiation can promote the apoptosis of malignant melanoma cells and inhibit their proliferation. This case was more suitable for heavy ions (^12^C^6+^). High-LET heavy-ion (C-ion) radiation (300 MeV/nucleon, 75 keV/μm) could significantly improve the ability to eliminate malignant melanoma cells by inducing apoptosis in tumor cells and inhibiting their proliferation in mice [367].

Proapoptotic Bax together with caspase-3 were significantly activated in hepatocellular carcinoma cell lines (HepG2 etc.) by C-ions (270 MeV/u and 13.3 keV/μm) [378]. In addition, in radioresistant human melanoma HTB140 cells, C-ion beams (LET: 197 and 382 keV/μm) significantly induced G2/M cell cycle arrest together with increases in the Bax/Bcl-2 ratio at 8–16 Gy from 1.24 to 3.38 and 1.65 to 2.52, respectively [370]. Significant increases in PARP cleavage were detected as well. Comparisons between X-ray and C-ion exposures of breast cancer cell lines (MDA-MP-23p, MCF-7) revealed a dose-dependent decrease in cyclin B1 and antiapoptotic Bcl-2 together with a strikingly more important G2/M arrest and marked increase in proapoptotic Bax activation [292]. Experiments in vivo on malignant melanoma tumor-bearing mice showed a significant up-regulation of proapoptotic Bax and cytochrome c (with down-regulation of Bcl-2 and proliferating cell nuclear antigen (PCNA)) [367]. These data clearly indicate that up-regulation of Bax by heavy-ion exposure (C-ions) is much stronger than by low-LET X-rays.

(3)IR-induced MOMPs.

MOMPs also initiate mitochondria-mediated apoptosis. In radioresistant cancer cells (p53 mutated or nonfunctional), low-LET IR-induced MOMPs, together with a slight increase in Bax, cytochrome c release and apoptosis, whereas high-LET heavy-ion irradiation with C-ions’ irradiation induced MOMPs more efficiently and intrinsic apoptosis including a significant increase in Bax levels and cytochrome c release [360].

(4)IR-induced effects on ATP production

Following exposures to 10 Gy of X-rays, human lung cancer A549 cells showed increases in mitochondrial ROS and ATP production [177]. In contrast, in a mouse model, high-LET carbon-ion irradiation (4 Gy) caused neural cell death in the hippocampus together with reduced mitochondrial integrity, disruption of TCA cycle flux and the ETC, depression of antioxidant defenses, persistent oxidative damage and significant decline in ATP production [379]. Interestingly, in human lung cancer A549 cells 2 Gy of C-ion irradiation considerably decreased ATP production [377]. From this, it appears that increased ATP production seen after low-LET X-irradiation may well support nDNA damage responses (DDR), whereas high-LET carbon-ion irradiation is associated with a significant decline in ATP production.

(5)Mitochondrial fragmentation

During IR-induced apoptosis, changes in mitochondria dynamics have also been noted. Jin et al., showed that X-rays and C-ions strikingly differ in the induction of mitochondrial fragmentation and apoptosis in HeLa cells [346]. With C-ion exposure, fragmentation was associated with apoptotic cell death and mitochondrial fission, whereas at low X-ray doses (0.5 Gy) mitochondrial fusion was observed, and ERK1/2-mediated phosphorylation of DRP1 on serine 616 mitochondrial fission was induced at higher doses (>1 Gy). Inhibition of mitochondrial fragmentation suppressed radiation-induced apoptosis for both types of irradiation. High doses (3 Gy) of C-ion (75 keV/μm) irradiation very efficiently induced mitochondrial fragmentation, and increased DRP1 and MFN1 expression (involved in mitochondrial fusion) was observed in a dose- and LET-dependent manner; cytochrome c was released from mitochondria and apoptosis occurred [346].

Moreover, using cytoplasmic single-particle irradiation with 3.4 MeV protons, Wang and Konishi showed that cytoplasmic irradiation promoted mitochondrial fragmentation, mitochondrial superoxide production, DSB induction, Nrf2 localization to the nucleus and down-regulation of p53 [380]. The Nrf2–ARE pathway is involved in a concerted action with the MRN complex (during DSB repair), HMGB1 and inflammatory cytokines [381]. This appears to indicate that IR-induced mitochondria fragmentation may promote responses beyond apoptosis, i.e., immune responses.

(6)IR induction of caspase activity and PARP cleavage

Activation of caspases is an important step in IR-induced mitochondria-mediated apoptosis. Importantly, during apoptosis, caspase activation is differentially modulated by IR depending on radiation quality. In general, the intrinsic apoptotic pathway is predominant and induction of apoptosis via caspase-9 and caspase-3, -6 and -7 is generally higher with high-LET IR than with low-LET IR. For instance, compared with X-rays, C-ions (13–100 keV/μm) induced apoptosis more efficiently in p53 mutated (radioresistant) human gingival cells (Ca9–22) via caspase-9 activation rather than caspase-8 activation [360]. Treatments with ^56^Fe ions (200 keV/μm) had effects that were comparable to C-ions. Both exposures lead to caspase-9 and subsequent caspase-3 and caspase-6 activation, and thereafter, to PARP1 cleavage, indicating the involvement of the mitochondrial intrinsic apoptotic pathway. Maximal apoptosis was observed at 100 keV/μm with C-ions and at 200 keV/μm for ^56^Fe ions. In addition, at an equivalent dose to that of X-rays (1.5 keV/μm), C-ions of 13 keV/μm were clearly more effective in cleaving PARP1.

Furthermore, after C-ion irradiation of human malignant glioma cells (at 2 Gy), Zhang et al., observed up-regulation of PARP and translocation of AIF translocation from mitochondria to the nucleus as well as induction of caspase-independent apoptosis accompanied by higher oxidative damage in mtDNA than in nDNA [382]. They suggested that heavy-ion (C-ion)-induced apoptotic pathways may be more activated by mtDNA damage (and/or by plasma membrane death receptors) than by nDNA damage.

Nakagawa et al., confirmed the high efficacy of ^56^Fe ion exposures (200 keV/μm) for the induction of apoptosis through activation of caspase-9 and caspase-3 at 2 Gy with an RBE of 3.5. ^56^Fe ions decreased the activity of serine/threonine-protein kinase B (Akt) [364]. At equitoxic doses, ^56^Fe ions were about eight times more effective than X-irradiation for the induction of intrinsic apoptosis. In accord with this, ^56^Fe ions induced G2/M cell cycle arrest in G2/M and cell cycle progression much more efficiently than X-rays [364].

In this context, exposures to 1 Gy ^56^Fe (180 keV/μm) and boron (200 keV/μm) heavy ions led to ROS induction associated with apoptosis and genomic instability involving TGFβ/Smad signaling in human fibroblasts and prostate cancer cells (PC8) with higher levels of Smad foci detected in micronuclei (MN) and evidence for TGFβ/Smad signaling at 24 h post-irradiation [383]. Induction of γH2AX and pATM (phosphorylated at Ser1981) was lower than that of Smad7 foci (colocalized to DDR proteins).

Following X-irradiation, the cytoprotective MAPK/ERK pathway with p38 and JNK was also activated [384] but not with C-ions (290 keV/μm) [285]. Indeed, Mori et al., had also observed that during high-LET-induced apoptosis, caspase-9 was phosphorylated at Tyr153 by MAP kinases such as ERK2 in a MEK-dependent manner [355]. Apparently, there are subtle differences in the metabolic routes that are activated during low- and high-LET-induced apoptosis.

(7)Involvement of SMAC in IR-induced apoptosis

According to Zhao et al., the role of the SMAC/direct inhibitor of apoptosis protein (IAP)-binding protein with low pI (DIABLO) protein also needs to be considered [385]. It is an essential and endogenous antagonist of IAPs, which is modulated in several cancers and can promote caspase activation. Smac/DIABLO was shown to interact with IAP proteins leading to increased caspase-3 activation and apoptosis [386]. In this way, Smac/DIABLO could sensitize TRAIL-resistant breast cancer cells in RT.

As stated by Zhang et al. [387], C-ion exposure significantly up-regulated the gene APOL6, known as a proapoptotic Bcl-2 homology 3-only protein inducing mitochondria-mediated apoptosis in cancer cells with the release of cytochrome c and Smac/DIABLO and activation of caspase-9 [388]. Importantly, the capacity of Smac/DIABLO for suppressing the apoptosis inhibitor protein AIP, freeing AIP and enhancing caspase activation is higher after high-LET IR than after low-LET exposure and results in strong caspase-9, -3 and -7 activation [385]. In fact, Smac/DIABLO antagonists (mimetics) have been proposed to increase the efficacy of anticancer therapy [385].

(8)IR-induced release of AIF, endoG and CAD

During apoptosis, cytochrome c release is associated with a decline in ATP production, loss of mitochondrial functions, Smac release, activation of mitochondrial AIF with chromatin condensation and endoG release with DNA degradation [389].

The release of mitochondrial endonucleases AIF and endoG-cleaving nDNA was higher with high-LET IR (leading to a more definitive cell degradation) than with low-LET IR [292,390]. These nucleases are especially active after high-LET particle irradiation. EndoG, an apoptotic DNase released from mitochondria [391], is part of the mitochondrial-directed apoptotic sequence of MOM permeabilization, release of cytochrome c, Smac/DIABLO, AIF and destruction by endoG and caspase-activated DNAse (CAD) nucleases [392,393].

Moreover, the induction of CAD-destroying DNA and cleaving PARP1 was more enhanced after high-LET IR (C-ion radiotherapy, CIRT) than after low-LET IR.

Curiously, in a human breast epithelial cell (MCF10A) population after exposure to ^56^Fe ions (600 MeV/u), some cells could survive up to 3 Gy despite caspase-3 activation, which normally initiates apoptosis. In the surviving cells, increased caspase-3 activation was associated with persistent DSBs involving the action of endonuclease endoG (after its translocation from mitochondria to nuclei for DNA fragmentation) and induction of chromosome aberrations [394,395].

In addition, Pandya et al., pointed out that expression of the Bcl-2 interacting killer (BikIK) protein, a stress-induced BH3-only protein, could activate caspases and genomic damage (DSBs) through CAD (and probably, also endoG) promoting aggressive cancer cell proliferation and decreasing cancer patient survival [396]. Thus, instead of fulminant apoptosis, BikIK could also induce sublethal apoptosis with limited caspase induction and increased induction of DSBs giving rise to cells with increased motility and stemness (invasion and migration capacity) as well as increased cancer cell proliferation.

#### 3.4.1. Partial Circumvention of IR-Induced Cell Death by Anastasis

Until quite recently, apoptosis has been considered to be an irreversible process [397]. However, recently, the phenomenon of anastasis has been described.

Tang et al., showed that in some conditions, cells can escape apoptosis [398]. After starting apoptosis, some cells can undergo Bax, cytochrome c release and caspase-3 activation but regain survival by a process called anastasis. Each step of mitochondria-mediated apoptosis may be subject to reversal, except the last one involving the final destruction of cell components by AIF, CAD, endoG and proteases. Tait et al., had noticed that after initiation of apoptosis not all mitochondria underwent MOMP [399], and the remaining healthy mitochondria could fuse and constitute a source of mitochondrial growth and repopulation that allowed cells to promote cell survival and proliferation and to resuscitate by anastasis [400,401]. Damaged (unhealthy) mitochondria were removed by mitophagy/autophagy-related proteins such as Atg12 and adapter proteins such as Sqstm1 [402,403]. Incomplete MOMP depending on cell type and caspase activities could initiate the up-regulation of genes related to posttranscriptional activities such as noncoding RNA processing, ribosome biogenesis, focal adhesion and cytoskeleton regulation.

MOMP-induced clonogenic death by IR was found to be mainly due to progressive loss of mitochondrial function and less dependent on the release of cytotoxic proteins from damaged mitochondria [390]. As shown by Tait et al., the ability of cells to survive MOMP depended on a few mitochondria that can evade permeabilization and repopulate cells [399].

Kaur et al., found that in human glioblastoma cells, IR could form polyploid giant cancer cells (PGCCs) with homotypic fusions that were associated with cancer cell survival and disease recurrence [404].

Incomplete or limited MOMP induced in a minority of mitochondria triggered caspase-3-dependent DNA damage and genome instability with transformation and tumorigenesis [405].

Regarding single cancer cell growth and formation of giant cancer cells after low-LET IR exposure, Mirzayans et al., observed [406] that after polyploidization/multinucleation and growth factor release were secreted, cell fusions could give rise to hybrid cancer cells (formation of PGCCs within irradiated tumors) retaining viability and developing resistance [407]. They proposed the following model: after low-LET IR exposure, polyploidy and multinucleation, growth factor secretion, depolyploidization, neosis and sub-genomic transmission via the release of DNA lead to persisting cancer cell survival together with tumor repopulation. The process includes cell-to-cell fusions (between cancer cells or with leukocytes), EMT and, during apoptosis, the release of caspase-3-dependent pro-survival factors with anastasis. Thus, caspase-3 can play a prominent role not only in the execution of apoptosis but also in carcinogenesis, metastasis and therapy resistance. Caspase-3-mediated secretion of pro-survival factors such as prostaglandin E2 may be involved, and caspase-3 and DNase mediated genomic instability.

In addition, Zhao et al., reported that activation of caspases does not irreversibly lead to cell death, in contrast to the established apoptosis paradigm. Moreover, in many cancer cells, spontaneous, unprovoked activation of caspases is maintaining tumorigenesis and metastasis involving changes in MOMP and sublethal activation of the apoptotic cascade with mitochondrial leakage of cytochrome c, caspase-3 and CAD activation plus endoG nuclear translocation [408]. IR-induced DSBs’ triggering of ATM-dependent activation of transcription factors NF-κB and STAT3 driving tumor growth, and late-phase exosome biosynthesis and release (after caspase-3 cleavage of the pore-forming protein DFNA) may contribute to maintaining the viability of neighboring cells and promote anastasis [408]. The important role of caspase-3 was demonstrated by the fact that caspase-3 knockout considerably reduced IR-induced tumor repopulation by decreasing endoG release from mitochondria to nuclei and impairing the ATM/p53/Cox2/PGF_2_ pathway in non-small cell lung cancer [409].

Regarding radiation, the quality determines outcomes of IR exposure in terms of the presence or absence (inhibition) of anastasis in apoptosis. The phenomenon of anastasis is because damaged cells seek and take other secondary metabolic pathways trying to ensure cell survival. After low-LET photon exposures, some cells succeed in escaping from apoptosis by fission and fusion of mitochondria to reconstitute healthy mitochondria from partially damaged, dysfunctional mitochondria [408,409].

In contrast, as shown above (see the section on apoptosis induction, Table 1, Figure 1) heavy-ion irradiation, e.g., exposure to C-ion beams, usually results in a more definite type of apoptosis at least at high doses (several Gy). Apparently, after C-ion exposure, because of the high density of ionization tracks and increased oxidative cellular stress induced, fewer healthy mitochondria are available and many mitochondria are strongly affected and dysfunctional, and unable to keep up with the high disorganization of the functional mitochondrial network. Anastasis appears to occur much more frequently after low-LET than after high-LET irradiation, albeit low-dose high-LET exposure may yield the same results as low-LET irradiation.

High (therapeutic) doses of C-ion irradiation prevent cancer cells from escaping apoptosis and show very much reduced anastasis. This appears to be due to the high capacity of C-ions to induce apoptosis associated with a reduction in autophagy and mitophagy compared with low-LET IR. Thus, therapeutic doses of C-ion exposure appear to favor a definite type and mitochondria-driven cell death avoiding mitochondrial reconstitution and anastasis. The high efficacy for the induction of apoptosis also has important consequences on bystander, non-targeted abscopal effects as well as on innate and adaptive immune responses.

Interestingly, a sort of rescue effect also appears to exist in bystander and non-targeted abscopal effects. Indeed, a reciprocal bystander or rescue effect has been described by which unirradiated bystander cells assist irradiated cells to survive and proliferate [410,411]. Kong et al., noticed that after X-irradiation of HeLa cells there was induction of autophagy and IL-6 secretion in bystander cells allowing metabolic cooperation [412]. Apparently, PARP1 regulates the IR-induced rescue effect [413].

#### 3.4.2. Modulation of Cell Death through Autophagy/Mitophagy Induced by Low and High LET IR

Autophagy and mitophagy induced by IR may exert cytoprotective or cytoptoxic effects in vitro but elicit immunogenic cell death in vivo with a likely important bearing in RT [324]. The underlying mechanisms are not yet fully understood but are thought to involved DNA damage signaling but also signaling from damaged plasma membranes, mitochondria and ER. Low and high LET IR appear to differ in their capacity to induce autophagy or mitophagy as exemplified in Table 2.

#### 3.4.3. Outcomes of High-Dose Low-LET and High-LET IR Effects on Cell Survival: Differences in Efficacy

High doses of low-LET or high-LET IR severely affect cell survival. Their efficacy differs as a function of radiation quality. In recent years, it has become evident that the effects on mitochondria can determine cellular fates after IR. Here, the hypothesis is put forward that the usual higher biological effectiveness of high-LET irradiation, in particular, heavy ions (taking C-ions as an example), as compared with low-LET photons, may include particular effects of high-density IR on mitochondria of normal and, even more specifically, of cancer cells. In fact, as central power stations, mitochondria are directing metabolism and signaling after IR exposure. Because of their specific intrinsic vulnerability and reactivity, they condition biological outcomes. Apart from epigenetic changes, IR-induced outcomes include mitochondria-mediated ROS production, fragmentation and mtDNA release, initiation of apoptotic cell death, mitophagy, release of cyto- and chemokines and bystander and distant abscopal effects as well as innate and adaptive immune responses.

##### Confirmation of the High Biological Effectiveness of High-LET IR

As already mentioned in the introduction, high-LET IR generally shows a higher biological efficacy than low-LET IR. The radiation-induced biological effectiveness (RBE) for protons and heavy ions is generally higher than that of low-LET IR [414]. In fact, Hartfiel et al., found in human cancer cells RBE values of 1.15, 2.3 and 2.5 for protons (64.1–70 MeV/u and 6 keV/μm), carbon (122.4–136.9 MeV/u and 101 keV/μm) and oxygen particles (141.4–160.9 MeV/u and 154 keV/μm) radiation in comparison to photons [269]. The RBE of C-ions (108 keV/μm) for the induction of apoptosis in NSCs was recently estimated to be ≈4 [415], quite similar to the RBE values observed for radiosensitive and radioresistant tumor cells (≈3–4 RBE_D10_, 100–200 keV/μm) of C-ion beams [416,417].

### 3.5. Involvement of Mitochondria in Low- and High-LET IR-Induced Bystander and Non-Targeted Effects

The discovery of bystander effects changed the common paradigm that after IR exposure only directly irradiated cells are affected by radiation-induced damage. Nagasawa and Little were among the first to show that irradiated cells initiate a wide range of intercellular communication affecting nearby neighboring cells, i.e., bystander cells or even more distant cells in cell populations and tissues [109,110]. Intercellular communication between those cells involved direct cell–cell interaction through intercellular gap junctions and/or intercellular signaling the emission of intercellular messenger molecules such as short- and long-lived radicals, chemo- and cytokines and nucleotides through culture or body fluids. For example, it has been demonstrated that α-ray particles and X-irradiated cells could communicate with nearby or distant unirradiated cells and induce bystander and non-targeted effects including induction of damage in unirradiated normal and cancer cells yielding inflammatory responses (fibrosis), genomic instability (in normal cells) and/or cell death (in normal and cancer cells) [9]. These bystander effects comprised a wide range of non-targeted IR responses including mitochondrial responses (see for review [163,164]).

Bystander and non-targeted responses mainly originate from IR-induced nuclear and mitochondrial damage signaling. One can distinguish (1) bystander effects within or nearby the irradiated (target) volume, (2) distant (abscopal) NTE outside the irradiated (target) volume, and (3) so-called cohort effects, i.e., NTE within an irradiated target volume (e.g., tumors). Bystander effects (DNA damage) induced by α-rays in normal human skin and confluent skin fibroblast cultures could extend up to 1 mm in tissue and 3 mm distance from irradiated cells in cell populations in culture medium [418,419], respectively, whereas NTE could extend through body fluids over large distances in the body.

#### 3.5.1. Involvement of IR-Induced Mitochondria-Driven Responses in Bystander and NTE Effects

In HeLa cells, Tartier et al., demonstrated that cytoplasmic exposure to α-rays induced mitochondrial ROS that resulted in the late occurrence (3 h post-irradiation) of nuclear 53BP1 foci indicating the induction of nDNA (DSBs) damage by mitochondrial ROS in unirradiated neighboring cells [420]. The effect was dependent on mitochondria because cytoplasmic α-ray microbeam exposure of HeLa cells lacking mitochondria could not elicit 53BP1 foci (DSB induction) in co-cultured HeLa mitochondria-competent cells. In addition, Choi et al., found that IR-induced MN derived from nDNA damage was induced by ROS from dysfunctional mitochondria in neighboring cells [421]. Thus, mitochondrial ROS can critically affect nDNA in bystander cells.

The crucial role of mitochondria and ROS in IR-induced bystander effects was confirmed by several authors [23,422,423,424,425,426]. Cells with mitochondria producing reactive oxygen and nitrogen species (ROS and NOS) after α-particle exposure were clearly involved in extracellular signaling and bystander effects [427,428]. Mitochondria generated extracellular signals from irradiated cells to bystander cells, which led to gene mutations, MN formation and activation of DDR and ATM in bystander cells [428]. Another proof came from the conditioned medium from irradiated human–hamster hybrid A_L_ cells lacking mitochondria which were unable to produce DSBs, mutations and cell death in unirradiated human skin fibroblasts [429]. Furthermore, inhibition of mitochondrial respiratory chain complexes I, III and V before IR exposure significantly reduced the capability of conditioned medium of A_L_ mitochondria-competent cells to induce gene (CD59) mutations in bystander cells [429].

Leakage of ROS and cytokines from mitochondria could be observed after α-particle irradiation from irradiated cells but also after cytoplasmic irradiation in unirradiated bystander cells yielding DSBs and MN [10]. However, mouse embryonic cells lacking cytochrome c in the intermembrane space of mitochondria did not show radiation-induced bystander effects signals from irradiated cells and were unable to produce MN in bystander cells [425]. Therefore, as stated by a recent review of bystander effects, the “early bystander response is initiated by a mitochondria-dependent increase in ROS triggering a signaling cascade that increases DNA damage in bystander cells depending on the make-up of the recipient cells” [426].

#### 3.5.2. Involvement of IR-Induced Mitochondria-Mediated Apoptosis in Bystander and Non-targeted Effects

As observed by Furlong et al., late apoptotic signaling (24 h) was also involved in low-dose IR bystander effects (0.05 Gy as compared with 0.5 Gy of γ-rays including up-regulation of p53, Bax, Bcl-2, caspase-2 and -6, but down-regulation of caspase-3 and -7, and thus establishing a close link between mitochondria-mediated apoptosis and signaling in bystander effects [430]. Furthermore, Portess et al., convincingly showed that IR-induced bystander effects can lead to apoptosis in cancerous cells: Exposures of 208F rat fibroblasts to low-LET (γ-rays) or high-LET α-rays at very low doses (50 and 20 mGy, respectively) stimulated the selective removal of precancerous 208Fsrc3 transformed bystander cells in co-culture by apoptosis following free radical (ROS/NOS) and cytokine signaling from damaged mitochondria [431]. These results were probably due to the increased vulnerability caused by the increased mitochondrial activity and ROS production of cancerous cells. As noticed by Prise and O’Sullivan, the extracellular signaling molecules secreted from irradiated cells for bystander effects often included nitrogen(II) oxide (NO), TGFβ1, TNF-α, IL-1, IL-8 and others [432]. Importantly, radiation-induced bystander effects have been linked to hallmarks of cancer such as radioresistance, tumor immune evasion, genomic instability, deregulation of energetics, tumor-promoting inflammation and sustained proliferation signaling. Thus, they are likely to affect RT outcomes [433].

Bystander effects have also been observed with high-LET carbon ions [434,435,436,437,438,439]. For example, bystander effects in terms of MN induction in nonirradiated HMy2.CIR lymphocyte cells co-cultured with irradiated U937 cells by carbon ions were higher than those obtained with γ-irradiation of U937 cells [437].

Furthermore, a protective and radioadaptive bystander effect could also be elicited by a low dose (0.2 Gy) of 1 GeV protons (LET 1.25 keV/μm) in normal human fibroblasts against DNA damage caused by subsequent ^56^Fe ions (5 Gy) (LET: 151 keV/μm) [440]. Unirradiated cells co-cultured with the low-dose proton-irradiated cells were significantly protected against the high challenge dose of ^56^Fe.

IR-induced bystander effects also are mediated by mtDNA and RNA components in exosome-like vesicles [441,442]. Their cargo contains a complex package of nDNA and mtDNA, cytokines (TGFβ and IL-10), HMGB1, mRNA, miRNA, circRNA, lncRNA and prostaglandin E2 (PGE2), endosomal sorting complex required for transport (ESCRT) and heat shock proteins (HSP) [443].

MiRNAs are found in exosomes and are part of epigenetic control processes (Section 2.4). Some miRNAs are related to specific IR exposures (8 Gy) [444,445].

At present, there are not many reports on bystander effects and exosomes produced after high-LET irradiation [446,447,448]. Interestingly, Yu et al. [449] and He et al. [450] reported that in exosomes (nanovesicles) derived from the serum of prostate cancer patients after treatment with CIRT (C-ions LET: 33.7 keV/μm), 66 Gy equivalents in daily fractions of 2.75 Gy equivalents) nine specific miRNAs were identified among which expression levels of miR654-3p and miR-378-5p could be correlated with CIRT efficacy and thus could serve as biomarkers for antitumor treatment efficacy [449,450]. This suggests that such exosomes may well be involved in beneficial bystander/abscopal effects [451] and in anticancer RT [452]. Although it has not been in the scope of this review, we might add here some evidence that NTE and abscopal effects are involved in immune responses. For example, Demaria et al., demonstrated that inhibition of distant untreated tumors (abscopal effects) observed after IR involves the immune system because IR-induced abscopal effects were absent in immune-deficient mice [453]. Although many similarities exist between abscopal effects of C-ion beams and of conventional low-LET RT [454], it is of interest to consider that the effectiveness of carbon ions is associated with the release of tumor-specific antigens and a higher immunogenicity fostering local and abscopal antitumor immune responses [455,456,457,458,459,460]. In addition, several preclinical studies reported antitumor immunity and abscopal effects of carbon-ion exposure using in vivo murine tumor models [461].

Therefore, abscopal effects are of interest because they take an active part in mitochondria-mediated innate and adaptive immune responses. High-LET exposures, e.g., with C-ions beams, appear to be especially effective.

## 4. Involvement of Mitochondria in Innate and Adaptive Immune Responses after IR

There is good evidence that mitochondria are not only involved in bioenergetic and biosynthetic metabolism but also in innate and adaptive immunity and signaling [462,463]. Following IR exposures, mitochondria are very much involved in the regulation of innate and adaptive immunity [462,464].

As mentioned above [9,10,11,12], exposures to IR introduce damage to all cell constituents including nDNA. In mammalian cells, the induction of DNA damage in nDNA activates the DDR signaling pathway with a more or less important cell cycle arrest allowing DNA repair. If DNA repair is incomplete or absent, programmed cell death, in most cases the mitochondrial (intrinsic and extrinsic apoptotic) pathways are initiated. Failure of DNA repair in the G2/M phase results in persistent DNA fragmentation and DSBs and the release of nDNA fragments in the form of MN. On the other hand, the mitochondrial apoptotic pathway also includes IR-induced mitochondrial dysfunction and initiation of mitochondrial signaling and the intrinsic and/or extrinsic apoptotic cascade involving fragmentation and release of mtDNA as well as effectors such as cytochrome c into the cytosol.

As also stated in the excellent recent review by Storozynsky and Hitt [465], IR-induced DNA damage plays a very important role in IR-induced cGAS-STING-mediated immune responses to cancer. In fact, cytosolic mtDNA and also nDNA are recognized by cytosolic proteins, i.e., so-called inflammasomes belonging to the innate immune system [59]. The inflammasome NLRP3 appears to react preferentially with oxidized mtDNA, whereas the inflammasome AIM2 recognizes double-stranded nDNA [466] and nonoxidized large mtDNA fragments [59]. The nDNA fragments are thought to be associated with IR-induced MN in the G2/M phase, whereas the release of mtDNA (complete or fragmented) into the cytosol is thought to be associated with mitochondrial intrinsic apoptosis after IR exposure [312,467,468]. For conformational reasons, mtDNA is more easily recognized than nDNA [59,469]. Below a certain threshold dose of IR (12–18 Gy), the innate immune response is activated in cancer cells [470] including recognition of dsDNA fragments by cyclic GMP-AMP (cGAMP) synthase and the stimulator of interferon genes (STING) sensing cytosolic DNA and mediating the release of type-1 interferon (IFN-1) [471]. Thus, nDNA and mtDNA can promote the innate immune response [472]; however, mtDNA is more efficient [473].

IR-induced mitochondrial responses play a crucial role in innate immunogenic (antitumor) responses [474]. Furthermore, IR-induced cell death signaling impacts on the tumor microenvironment and innate immunogenic responses [475]. Anticancer immunity is thought to start with mitochondrial apoptosis as the initiating event, including the induction of MOMP, the release of cytochrome c, macropore formation in the mitochondrial outer membrane accompanied by the release of mtDNA into the cytosol, recognition by the cGAS-STING pathway and cGAMP synthase-driven IFN-1 secretion ensuring final immunogenic cancer cell death [474,476].

As shown in Figure 3, C-ions are usually more efficient in the induction of DNA damage, MOMP and apoptosis than low-LET IR. Induction of MOMP is known to trigger autophagy (and mitophagy) and diminish mtDNA release and INF production, whereas apoptosis leads to degradation of mitochondria and inhibition of INF secretion [477].

From this, it appears likely that the increased therapeutic effectiveness of high-LET C-ions in CIRT is due to their high effectiveness in inducing MOMP and apoptosis, and modulation of mitophagy promoting immunogenic antitumor responses. Obviously, the complex interplay between the specific effects of exposure to heavy ions (e.g., C-ions) on nuclear and mitochondrial damage and the immunogenic consequences merits further analysis (see Figure 4 below).

## 5. Discussion and Conclusions

The involvement of mitochondria in IR-induced responses may be schematically summarized as follows (Figure 4):

Biological responses to IR are not only dependent on genetic control but also on epigenetic control. It is well-known that IR affects genomic DNA in the first place. Unrepaired radiation damage in nDNA may lead to cell death, and incorrectly repaired nDNA damage may be genetically fixed, giving rise to mutations that are transmitted to the next generation of cells and tissues and lead to malignant cell transformation and cancer. Mechanistically, DDR signaling is crucial for relatively early genetic responses and direct long-term consequences. However, as emphasized here after IR exposure, the epigenetic control by mitochondria is also very important.

In recent years, it has been increasingly recognized that mitochondria play an essential role in cell metabolism and signaling but also epigenetic control and inheritance [9,16,78]. The involvement of mitochondria in epigenetic alterations of nDNA, mtDNA and proteins (histones) confirms its central role in cellular homeostasis and reaction dynamics. Very importantly, they impact on NTE, abscopal and innate and adaptive immune responses after IR and thus are decisive determinants of long-term outcomes in low-LET RT and high-LET RT (CIRT).

This review shows that mitochondria are not only central power stations for metabolism and energy supply but also crucial signaling elements for the fine-tuning of cellular, tissue and whole-body radiation responses. In addition, it emphasizes that mitochondria are implicated in low-dose radiation effects such as low-dose adaptive responses, i.e., low-dose radioadaptation, low-dose hyperradiosensitivity, as well as NTE, genomic instability and carcinogenesis involving pro-cancer inflammatory and/or anticancer immunogenic effects.

The analysis of IR effects on mitochondria-mediated responses reveals hitherto as yet unrecognized extraordinary specificity of IR, particularly that of high-LET heavy ions such as carbon ions in directing molecular signaling pathways toward selective survival, cell death and immunogenic or immunosuppressive responses. Undoubtedly, this widens existing radiobiological concepts in radioprotection and anticancer radiotherapy.

In fact, the available data reported above are renewing and exceeding established views and appear to converge toward a unifying concept illustrated in Figure 4.

Depending on radiation quality, energy disposition IR induces damage to all cellular constituents on the molecular level including structural and functional changes in the nucleus, and cytoplasmic organelles such as mitochondria [12]. In general, high-LET IR has a stronger impact than low-LET IR [8].

Damage to nDNA gives rise to the DDR response with activation of molecular signaling toward DNA repair, cell cycle arrest, survival and/or cell death. Misrepair or the absence of repair may lead to mutations, genomic instability, senescence, malignant transformation and cancer and death [4].

Concomitantly, damage to mtDNA leads to mitochondrial dysfunction, altered bioenergetic and biosynthetic metabolism, generation of ROS and oxidative stress leading to additional damage to mtDNA and also nDNA, proteins and membrane lipids and mitochondria-mediated signaling of programmed cell death (intrinsic and/or extrinsic apoptosis), non-targeted and bystander effects, as well as important innate and adaptive immune responses [9].

For this, nuclear and mitochondrial functions are highly interconnected in the cellular network together with the ER (center of UPR) and the Golgi apparatus [21].

To summarize the involvement of mitochondria in IR-induced effects on mammalian (and human) cells, first comes to mind that energy supply by mitochondria through OXPHOS or glycolysis in the form of ATP is a very important regulator of IR responses including the DDR response, cell survival, growth and proliferation. Energy production involves leakage of mitochondrial ROS (mtROS). Usually, exposures to IR exacerbate mtROS production due to the damage inflicted to mitochondrial components (mtDNA, proteins and membranes) and alterations induced in the electron transfer chain complexes. The increase in ROS and oxidative stress affects irradiated and also bystander cells. As shown by Dettmering et al., C-ion irradiation is clearly more effective than low-LET IR in the production of ROS in normal human fibroblasts [179] but also in mtDNA and nDNA fragmentation [346,369].

### 5.1. Low-Dose Considerations

At low IR doses (<100 mGy), the ROS produced can be either stimulating and give rise to adaptive responses with increases in antioxidant enzymes, ATP production, mitochondrial dynamics (fission and fusion) and innate and adaptive immune responses [9,16]. Sometimes, low-dose nonlinear responses have been observed.

For example, low doses may be able to eliminate via apoptosis precancerous cells by radiation (α-rays)-induced bystander effects from irradiated normal cells [431]. They may also increase cellular defenses against pro-oxidant forces from environmental stress and sustain genomic integrity and longevity [478]. The effects of fractionated dose and chronic IR and low-dose-rate exposures need to be considered to be well [479].

For example, low-dose-rate effects have been observed concerning endothelial cell radiosensitivity [480], neoplastic transformation [481], tumor cell responses (*in vitro* and *in vivo*) [482] and even human tumor (prostate) responses to RT [483].

However, low-dose responses clearly differ in normal and cancer cells [484]. In fact, in quiescent cancer stem cells, low doses may as act “wake-up” signals [485,486]. C-ions and miRNAs may inhibit them [487]. This also indicates that low-dose IR stimulates cancer cells to produce a vicious circle of mitochondrial malfunction producing more mtROS-dependent damage and mediating persistent oxidative stress and genomic instability and carcinogenesis. A link might exist between IR-induced oxidative stress, epigenetic alterations and genomic instability. This implies that low doses of IR give rise to very variable results depending on the metabolic and physiological (oxidative) status of the cells, tissues, organs and organisms. Obviously, the responses in the low-dose range can be nonlinear depending on the system. Thus, cancer risk estimations using extrapolations from high down to low doses, assuming a strictly linear relationship according to the LNT hypothesis employed in radioprotection, find here important limitations [164,192,488,489].

In summary in this respect, one can note that although cellular damage appears to be induced in an LNT dose-dependent fashion, radiation doses are not always equally and proportionally damaging to the same extent over the whole dose range because of the specifically finely tuned adaptive reactivity of cellular and tissue networks of biological systems in response to the different amounts and types of damage inflicted and time-dependent manifestations. One reason the LNT model continues to be questioned [209,227,490,491] may well be because until now low-dose effects including epigenetic effects and mitochondrial functions have not been considered. Nonlinear effects were observed when comparing low- and high-dose IR effects [163,164] that appear to involve mitochondrial regulatory and signaling functions.

### 5.2. High-Dose Considerations

High doses (>1 Gy) often used in RT (also encountered in severe radiation accidents) are known to induce gene mutations and cancer but also cell death (apoptosis). In general, high-LET IR is significantly more effective than low-LET IR [479]. Often deletion mutations are induced in nDNA and mtDNA that can inactivate important tumor suppressor genes such as p53, Myc and Ras that can initiate carcinogenesis [9].

However, the capability of low- and high-dose IR to inflict significant damage simultaneously to all cellular constituents in an energy- and dose-dependent manner has made IR a very efficient tool for identifying subtle or important molecular targets, mechanisms and biological consequences. Moderate (0.1–1 Gy) and high doses (>1 Gy) of IR also launched the debate in radioprotection, how best healthy tissue can be protected from IR-induced damage and the relatively high doses (>1 Gy) served to seek best ways for treating cancers by RT.

As shown in this review, high doses of IR can affect mitochondrial function quite severely. These doses usually provoke important mitochondrial dysfunction and signaling that can either lead to partial apoptosis with some recovery (anastasis) in a subset of cells (low-LET photon IR) or to complete definite apoptosis (by high doses and high-LET IR (e.g., C-ions).

### 5.3. High-LET Considerations and Radiotherapeutic Outcomes

Regarding high-LET exposures (e.g., C-ions), their specificity appears to rely on two main features, (1) the precise targeting of tumors (including radioresistant ones, such as glioblastoma and pancreas) associated with a high local energy deposition, and (2) in the very high ability to induce mitochondrial dysfunction associated with non-irreversible apoptosis, i.e., inhibition of anastasis together with down-regulation of general and selective autophagy (mitophagy). Both contribute to the very high RBE (>3) typical for high-LET radiation. Furthermore, the high level of mitochondrial dysfunction generated, in part associated with the ER- and the UPR-stress responses, then leads very efficiently to abscopal and immunogenic responses limiting inflammatory immune effects. Some of these features can, at least partially, be found with other accelerated high-LET particles (α-particles and accelerated oxygen, silicon, neon and iron ions).

It is thought that the destructive power of highly dense ionization tracks causes mitochondrial dysfunction, metabolic distress and widespread ROS that can overwhelm even resistant cancer cell and cancer stem cell defenses. Apparently, these particles (C-ions) may leave some space for the activation of neighboring cells and especially, circulating active immune cells to react, recognize and eliminate aberrant cancer cells and inhibit tumor growth and metastasis formation.

This has important consequences for therapeutic outcomes of hadron therapy using CIRT. In fact, many clinical observations and trials on patients with radioresistant solid tumors (head and neck, glioblastoma, pancreas, etc.) revealed that CIRT yields more beneficial and favorable long-term outcomes, i.e., fewer side effects or less metastasis and secondary cancers [416,456]. In particular, radioresistant tumor types such as glioblastoma, chondrosarcoma and head and neck cancers, non-small cell lung cancers, hepatocellular carcinoma, colorectal and gastrointestinal cancers, sinonasal malignancies, prostate and gynecological cancers have been successfully treated [456,477,478,492,493,494,495,496,497,498,499,500]. For that the radiation quality of C-ions matters a lot.

In addition, the effects of C-ion irradiation can be increased further through inhibition of DDRs and immune checkpoints [501,502,503].

Interestingly, very recent data reveal that somewhat similar results may be obtained with conventional RT in combination with DNA repair and immune checkpoint inhibitors [501,504,505,506] and also with the new evolving technique using high intensity short FLASH IR exposures [507,508,509,510,511,512]. Moreover, these approaches were also able to attenuate inflammatory and fibrotic responses as well as the formation of metastasis and secondary cancers. Thus, it is of great interest to explore further the involvement of radiation-induced mitochondrial dysfunction in these favorable anticancer therapies employing low- and high-LET RTs including CIRT.

Herein, clear evidence is provided that depending on radiation quality and cell and tissue type, important radiosensitization can be achieved not only by modulating mitochondrial metabolism but also by mitochondria-mediated intra- and intercellular signaling and activation of the innate and adaptive immune systems.

## Figures and Tables

**Figure 1 ijms-22-11047-f001:**
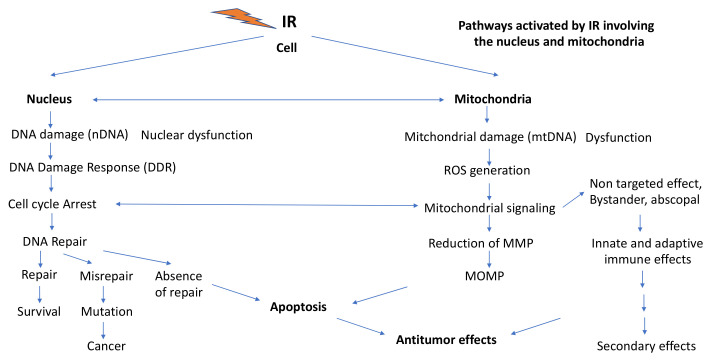
Pathways activated by ionizing radiation (IR) involving the nucleus and mitochondria including activation of the DNA damage response and mitochondrial signaling toward apoptosis and innate and adaptive immune responses with either antitumor effects or secondary effects (inflammation). Acronyms: ROS: Reactive oxygen species; MMP: Mitochondrial membrane potential; MOMP: Mitochondrial outer membrane permeabilization.

**Figure 2 ijms-22-11047-f002:**
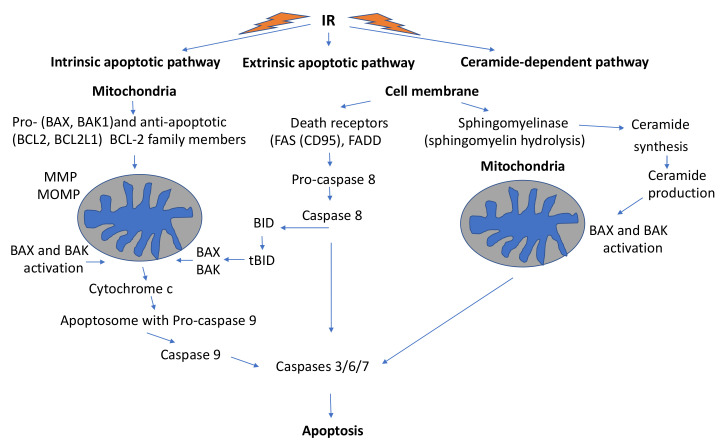
Main mitochondria-mediated cell death pathways induced by ionizing radiation (IR): Intrinsic apoptosis, extrinsic apoptosis and ceramide-dependent apoptosis. Acronyms: Bax: Bcl-2 associated protein; Bak: Bcl-2 homologous antagonist/killer; Bcl-2: B-cell lymphoma; Bcl-2L1: B-cell lymphoma associated ligand 1; FAS: Fas cell surface death receptor; FADD: Fas associated with death domain; MMP: Mitochondrial membrane potential; MOMP: Mitochondria outer membrane permeabilization; Bid: BH3 interacting domain death antagonist; tBid: truncated BID.

**Figure 3 ijms-22-11047-f003:**
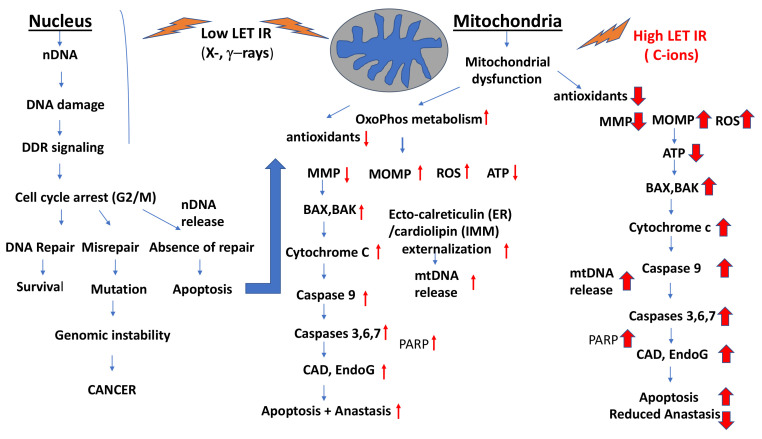
Differential effects of low-LET and high LET (C-ions) radiation on the induction of damage to the nucleus and to mitochondria resulting in apoptosis with or without anastasis, respectively. Small red arrows indicate moderate up–or down-regulation after low-LET exposure, whereas the big red arrows indicate very strong up-or down-regulation after high LET (C-ion) exposure. Acronyms: DDR: DNA Damage Response; ATP: adenosine triphosphate, ROS: reactive oxygen species; Bax: Bcl-2 associated protein; Bak: Bcl-2 homologous antagonist/killer; IMM: inner mitochondrial matrix, ER: endoplasmic reticulum; MMP: Mitochondrial membrane potential; MOMP: Mitochondria outer membrane permeabilization; PARP: Poly (ADP-ribosyl) polymerase, CAD: Caspase-dependent DNase; EndoG: endonuclease G.

**Figure 4 ijms-22-11047-f004:**
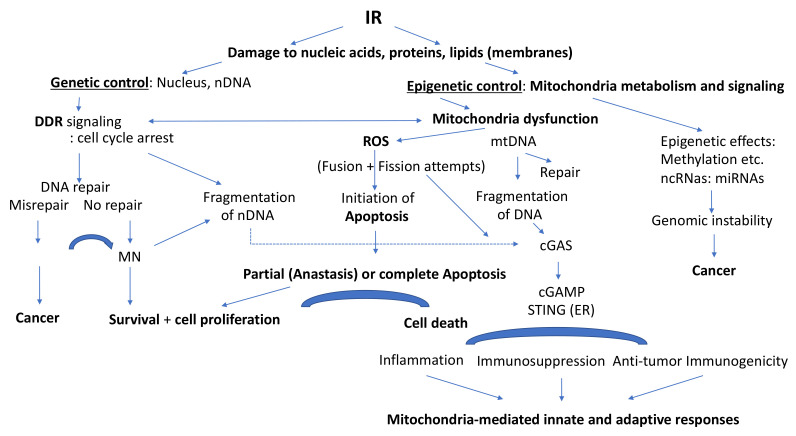
General scheme of intermingling of genetic control and epigenetic control involving the nucleus and mitochondrial metabolism and signaling. Ionizing radiation (IR) induces DNA Damage Response (DDR) and mitochondrial dysfunction promoting partial (anastasis) or complete apoptosis, and mitochondria-mediated innate and adaptive immune responses which may exert anti-tumor responses. IR-induced cancers may arise either from misrepair or absence of DNA repair, or from IR-mediated interference with mitochondria-dependent epigenetic control. Acronyms: cGAS: cyclic GAMP synthase; cGAMP: cyclic GMP-AMP; STING: stimulator of interferon genes.

**Table 1 ijms-22-11047-t001:** Induction of apoptosis by low-LET IR and/or high LET IR.

Radiation	Cell System	Apoptosis	References
γ-rays/C-ions (105 keV/μm)	Glioblastoma cells (p53 wt/p53mutant)	C-ions are more effective than γ-rays. p53 mutants are more resistant than p53wt, higher G2/M block in p53mutants than in p53wt.	[353]
γ-rays/C-ions (20–80 keV/μm) Dose: 10 Gy	Glioblastoma cells (U87MG, p53 wt) and normal human fibroblasts (FB) (NB1RGB)	High induction of apoptosis by C-ions in normal and especially in glioblastoma cells. Fibroblasts show less apoptosis than glioblastoma cells.	[361]
X-rays/C-ions (13–100 keV/μm)/56Fe ions (200 keV/μm)	Human gingival cells (Ca9-99)	C-ions and 56Fe ions were more effective than X-rays in inducing apoptosis.	[360]
X-rays/C-ions (33.4 and 184 keV/μm)	Head and neck squamous cell carcinoma cells (HNSCC): SCC61 (radiosensitive) and SQ20B(radioresistant)	Induction of early (p53-independent) apoptosis: higher effectiveness of C-ions than X-rays.	[363]
γ-rays/C-ions (290 keV/μm)	Lung adenocarcinoma A549 cells	Early and more effective apoptosis induction by C-ions compared to γ-rays. C-ions induced predominantly pro-apoptotic signals: JNK transiently, but not ERK1, although ERK1 was activated by γ-rays (1 Gy).	[285]
X-rays/56Fe ions (200 keV/μm)	Human gingival cancer cells (Ca9-22)	56Fe ions were about 8 times more effective than X-rays in apoptosis induction	[364]
Photons/C-ions (103 keV/μm)	Human glioblastoma (U87)	Higher efficacy of C-ions than photons (RBE: 3.3–3.9)	[365]
X-rays/C-ions (33.6 and 184 keV/μm) Dose: 10Gy	Head and neck squamous cell carcinoma cells (HNSCC): SCC61 (radiosensitive) and SQ20B (radioresistant)	Late apoptosis > 120 h post-irradiation was independent of p53 status, higher after C-ions than after X-rays, involving ceramide synthesis	[322]
X-rays/C-ions (keV103/μm)	Human cancer cells (A549, LN-229, PANC-1)	C-ions much more effective than photons for induction of apoptosis	[366]
X-rays/C-ions (33.6 keV/μm)	Human glioblastoma cell lines	Ceramide-dependent late apoptosis, higher after C-ions than after X-rays	[356]
X-rays/C-ions	Prostate cancer cells (PC3)	Induction of a persistent G2/M cell cycle arrest with C-ions	[291]
Low-LET photons/^12^C-ions/^16^O-ions	Hepatocarcinoma cell lines	Increased growth arrest (RBE: 1.9–3.3)	[293]
Low-LET photons/C-ions	Mice: malignant melanoma cells BF16F10 and tumors	C-ions very effectively increased apoptosis and inhibited cell proliferation. In vivo, tumor growth was inhibited by C-ions to 95% and by X-rays to 37.5%.	[367]
50 kVp X-rays/C-ions (75 keV/μm) Dose: 2 Gy	Mouse sarcoma 180 cells	C-ions were significantly more effective than X-rays in inducing intrinsic apoptosis in vitro and in vivo.	[368]
C-ions (75keV/μm) Doses: 0.5 and 3 Gy	Human breast cancer cells (MDA-MB-231)	Considerable increase in LC3 expression (autophagy), cleaved caspase-3 expression and strong apoptosis (strong cell killing)	[346]
X-rays/C-ions (70 keV/μm)	Human cervical cancer cells (HeLa)	Significantly increased cytochrome c release and apoptosis at 3 Gy; both could be inhibited by an inhibitor of cytochrome c release (Mivi-1)	[369]
X-rays/^12^C-ions (101 keV/μm), ^16^O -ions (141 keV/μm)	Human malignant melanoma cell lines WM115, WM266-4 (metastatic)	Photon RT (2–6 Gy) resulted in G2/M arrests comparable to 0.5–2 Gy with ^12^C-and ^16^O-ions. Heavy ions were up to 4 times more effective than RT photons.	[362]

**Table 2 ijms-22-11047-t002:** Autophagy/mitophagy following low or high LET IR.

Radiation	Cell System	Autophagy/Mitophagy	Reference
IR	Mammalian cells	Induction of auto- phagy	[9]
Low-LET IR	Mammalian cells	Regulation of autophagy by mTOR/complex Akt/PI3K can be blocked by IR	[296]
Low-LET IR	Mammalian cells	The DDR proteins p53, ATM, PARP1, FOXO3a, mTOR and SIRT1 are involved in the regulation of autophagy.	[324]
Low-LET IR	Mammalian cells	mTORC1 activity is induced by low levels of ROS, inhibited by higher ROS levels and mitochondria mass reduced via mitophagy	[41]
C-ions (75 keV/μm) Dose: 2 Gy	Sarcoma cell line S180	High ER stress with subsequent IRE1 signaling, autophagy (via IRE1/INK/p-Bcl-2/Beclin-1 signaling axis) and apoptosis. Apoptosis could be further increased with chloroquine 10 μg 4 h treatment	[368]
X-rays/C-ions (75 keV/μm)	Human cervical cancer cells (HeLa)	Mitochondrial fusion occurred at low dose X-rays (0.5 Gy) and ERK1/2 -mediated phosphorylation of DRP1 at higher doses (> 1 Gy Gy). C-ions induce effective mitochondria fragmentation, fusion (increased DRP and MFN expression) and apoptosis	[369]
C-ions (75 keV/μm)	Human breast cancer cells (MCF-7, MDA-MB-231)	At low dose (0.5 Gy) of C-ions damaged mitochondria were eliminated by mitophagy. High-dose 3 Gy exposure resulted in high BAX expression, cytochrome release and apoptosis. Autophagy inhibition led to mitophagy and increased survival.	[346]
Protons (microbeam)	Normal human lung fibroblast WI-38 cells (Cytoplasmic irradiation)	Mitochondrial fragmentation, induction of ROS and DSBs, localization of NRF to the nucleus, down-regulation of p53.	[380]
C-ions 300 MeV/u Dose: 4 Gy	Mouse hippocampus	Changes in LC3II/LC3I ratio, co-localization of LC3 with COX IV (indicative for the presence of mitophagosomes), PINK1 and Parkin levels were markedly reduced;overexpression of NRF2 and PINK1 in mouse hippocampus HT122 cells and apoptosis attenuated.	[379]
Low-LET IR (γ-rays)	Human normal fetal lung fibroblasts	Mitophagy involves the Spata 18 pathway and accumulation of lysosome-like organelles in mitochondria	[342]
Proton microbeam	Human lung fibroblasts (WI-38)	Increased ROS, DSB induction, Nrf2 activation, mitochondria fragmentation and down-regulation of p53	[380]
X-rays (10 mGy, 50 mGy fractions, total doses 460 mGy, 2.3 Gy): low dose long-term exposures (31 days)	Human fibroblasts, ATM (+/+), ATM-defective (AT5BIVA), NBS1+/+, NBS -defective (GM7166)	Low-dose long-term X-irradiation increased mitochondrial mass and ROS in ATM (+/+) and NBS(+/+) cells leading to mitophagy involving Parkin1. In ATM and NBS-defective cells defective mitophagy was induced leading to mitochondrial fragmentation, decreased mitochondrial membrane potential and apoptosis.	[345]

## Data Availability

The data presented are taken from the scientific literature as referenced (Pubmed).

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
