# Peer review of "Role of Mitochondria in Radiation Responses: Epigenetic, Metabolic, and Signaling Impacts"

_ijms, 2021, doi:10.3390/ijms222011047_

Round 1

Reviewer 1 Report

Many thank for your manuscript. It contains a vast amount of information on mitochondria and their role in the effects of ionizing radiation exposure in normal and cancerous cells. What I lack in the review is some sort of overview in the beginning, a content table, to have a quick overview of the manuscript. Furthermore, there is a lack of consistency in abbreviation, word and bracket use as well as font size (e.g. C-ions and 12C ions; abbreviation of IR explained numerous times in the manuscript; bigger font at lines 1100 and 1101). For some sections there is a lack of mitochondrial-specific information (e.g. 2.4.2). At line 1373 it reads "heavy ions and C-ions". C-ions are a part of heavy ions as a group. Please correct. When discussing the high dose considerations (section 5.2 starting at line 2273) there seems to be an issue with the dose definition. I think you meant to say > 1Gy. Furthermore, on line 2283 you say "relative" high doses. What is the line between high doses and relative high doses? Most of my remarks can be found in the annexed pdf-file of your manuscript.

Author Response

Response to reviewer 1 :

Dear reviewer,

We deeply acknowledge the comments of reviewer 1. All suggestions were well taken and the text corrected accordingly.

Details:

Lines corrected in the original version:

44: corrected IR

46: corrected O2

51: radiation

72: used in radiotherapy

237: TFAM

297: specified: in genomic DNA

401: letter size corrected

494: spDSB replaced by “They”

782-784: letter size corrected

790-792: letter size corrected

796   now 793: space corrected

815-820 now 812-816: letter size corrected

828-831 now 823-826: letter size corrected

 Change in Figure 1: cellule and RT deleted

1098 now 1093: C-ions

1098 now 1094: C-ions, letter size corrected

1315- 1316 now 1310-1311: letter size corrected

1373 now 1368: “heavy ions and C-ions” deleted

1797: paragraph heading completed according to editor suggestion:  3.5.1.1.

2273 now 2269: > 1Gy

2283 now 2279: “relative” and “(> 1Gy)” deleted

Please, note that the finalized new version not only contains your corrections but also those of the other reviewers. You will notice that the whole manuscript underwent quite a substantial shortening:  two chapters 3.4.2 , 3.5.2, 3.5.3 and chapter 4 and chapter 5 were shortened according to reviewers’ request and tables 3 and 4 deleted together with about a hundred references. In spite of this, we believe that the main messages of the paper are still

kept.

Thanking you again for reviewing the paper and your kind input,

With best regards,

Dr. Dietrich Averbeck

Reviewer 2 Report

In the manuscript presented by Averbeck and Rodriguez-Lafrasse, authors aimed to review the current knowledge on the impact of mitochondria-dependent epigenetic and functional control at the cellular radiation response. In more detail, they indicate that low dose radiation effects on mitochondria appear to be associated with epigenetic and non-targeted effects, genomic instability and adaptive responses, while high dose therapeutic effects may involve mitochondria-mediated innate and adaptive immune responses. Moreover, special emphasis is given on effects of radiation quality comparing low and high LET exposures.

Authors put very high efforts in writing the manuscript and selecting references to perform a very comprehensive review on up-to-date and interesting issues in radiation biology. Overall, the review is well written and comprises an excellent compilation of the most relevant references on the topic and thus, may allow the reader to gain access to current findings and concepts. The manuscript, however, is extraordinarily complex and somehow exhausting the reader while follow the huge amount of information provided. Authors are encouraged to shorten the manuscript by omitting textbook knowledge on DDR and a detailed review on immunologic effects of low and high LET exposure, avoiding redundancies. Major suggestions of improvement are given successive.

Major points of improvement:

  1. Paragraph “3.2.1. Hormesis” comprehensively reviews findings associated with the cellular response to low dose irradiation (lines 716- 854), but only indirectly provides evidences on the impact of mitochondria.
  2. In line with that, lines 1011-1035 and 1047-1075 mainly cover textbook knowledge on DNA damage induction/repair and the LET effect and thus, should be shortened.
  3. Figure 1. Authors nicely reviewed on a multitude of mechanisms of IR induced cell death (paragraph 3.3.3). Accordingly, apoptosis should be replaced by cell death. In addition, acronyms provided in the figures should be given in the legend to the figure (also in Figure 2).
  4. Figure 2 is misleading in terms of pro caspase 8 activation that is not associated with the mitochondrion in a direct manner. Authors in addition should include BID in the cartoon.
  5. Paragraph “3.4.2. Outcomes of high dose Low LET and high LET IR effects on cell survival”, lines 1702-1721 are somehow redundant to the former chapters.
  6. Paragraph “3.5.2. Role of radiation quality in bystander and NTEs” as well as “3.5.3. “Involvement of exosomes in bystander and NTEs” do not or to a low extent include findings on mitochondrial physiology and accordingly do not contribute to the major purpose of the review article. The paragraph should be shortened/omitted.
  7. In addition, the paragraph “3.5.4. Abscopal effects (NTEs) and immune effects” is completely out of the scope of the review article. The same holds true for paragraph 4.2.1: “Association of immunogenic effects with apoptosis” including table 3 and table 4, exclusively focusing on immunogenic effects of low and high LET irradiation. These data/findings should be subjected to an independent review article.

Author Response

Response to reviewer 2  240921

Dear reviewer 2,

We are very grateful for your very pertinent suggestions and appreciated your points of improvement. Thanks to you we revised the manuscript quite substantially: it has been drastically shortened, Figures 1 and 2 were revised and tables 3 and 4 deleted and the text trimmed down accordingly.

Details of revision:

Point 1:

Paragraph “3.2.1. Hormesis”: we focus now on hormesis in relation to mitochondrial metabolism:

Lines 743-748: deleted with refs. 197, 198

           754-766: deleted with refs. 188, 203-205

           784-787 deleted with refs. 215-216

           807- 831 deleted with refs. 217-221, 222-227

           912-920 deleted with refs. 219-218

On line 501, we inserted a sentence “ The whole metabolic and signaling network are involved as well as mitochondria, at least indirectly; However, case-specific predictive biomarkers are still missing”.

Point 2:

 Lines 1014-1021 deleted with refs. 295-299

            1024-1033 deleted with refs. 305-324

Point 3: Figure 1

               We provide acronyms in the legends (also in Figure 2) and reviewed chapter 3.3.3.

                We replaced in line 1115: apoptosis by “cell death”.

Point 4: We redraw Figure 2 according to the reviewers’ suggestion (and introduced BID)

Point 5: chapter 3.4.2: we deleted mines 1703-1721

Point 6: chapter 3.5.2. was drastically shortened: lines 1821-1838 were deleted but refs. kept. Chapter 3.5.3. was severely trimmed down: lines 1839- 1849 deleted. We added a sentence after line 1889-1920 and deleted refs.500-507

Point 7: The chapters 3.5.4.  (abscopal, NTE and immunogenic effects) and 4.2.1 now nearly disappeared completely after important shortenings and deletion of refs. 522-523, 525-526 and of Tables 3 and 4, as suggested.

Corrections can be found on the Ijms-13594 editor’s version 040921c1 rev2 dac.pdf

A corrected version including the corrections according to all reviewers can be found in the

newly submitted version of the manuscript.

We believe that the manuscript has now very much improved.

With many thanks for your very helpful suggestions,

with best regards,

Dr. Dietrich Averbeck

Reviewer 3 Report

Please define LET and CIRT in the abstract.

The review is too long and an excessive number of references is provided, the text needs to be condensed and the number of references to be decreased.

line 78: ETC is a part of OXPHOS, not two different things, please rephrase

OXPHOS should be designated the same way throughout the manuscript.

The manuscript would benefit from a Figure indicating basic mitochondrial functions and characteristics as Figure 1.

The authors discuss heavily cancer cells, although this is not mentioned in the abstract/introduction as the focus of the manuscript. Please modify accordingly.

Author Response

Response to reviewer 3

Dear reviewer 3,

We very much appreciated your comments and suggestions concerning the improvement of our manuscript. All your points were well taken.

  • We defined changed the wording in the abstract concerning LET and CIRT.
  • We shorted the paper and deleted about hundred references and condensed the chapters on hormesis, bystander, NTEs and immunogenic effects ( also asked for by reviewer 2).
  • We avoid mixing ETC with OXPHOS and replaced ETC by OXPHOS throughout the paper according to your suggestion.
  • Indeed, the introduction of a figure 1 indicating basic functions and characteristics of mitochondria is a very good idea. However, because of the desire of reviewer 2 to shorten the manuscript we refrained from adding another figure.
  • We mention now in the abstract that we focus more on cancer cells than on normal cells.

Please, find the corrected version in ijms-1359474 editor’s version 240921 rev 3, and the finalized version containing the corrections of all reviewers in the newly submitted version.

With many thanks again for your very useful suggestions which helped improving our paper,

with best regards,

Dr. Dietrich Averbeck

Round 2

Reviewer 2 Report

In a revised version of the manuscript, the authors addressed the previous concerns in an convincing manner.

Reviewer 3 Report

Please provide a version where all changes made are tracked, as currently it is not possible to see all changes made in the text.

Also, I still believe that the manuscript would benefit from the suggested Figure 1.

Round 3

Reviewer 3 Report

no additional comments